# Utilization of Waste Glass Cullet as Partial Substitutions of Coarse Aggregate to Produce Eco-Friendly Concrete: Role of Metakaolin as Cement Replacement

**Noor Md. Sadiqul Hasan** [1] , **Nur Mohammad Nazmus Shaurdho** [1] , **Md. Habibur Rahman Sobuz** [2] ,
**Md. Montaseer Meraz** [2] , **Md. Saidul Islam** [1] **and Md Jihad Miah** [3,*]

1 Department of Civil Engineering, International University of Business Agriculture and Technology, Dhaka 1230, Bangladesh; nmshasan@iubat.edu (N.M.S.H.); nmnshaurdho@gmail.com (N.M.N.S.); saidul.bappy1990@gmail.com (M.S.I.)
2 Department of Building Engineering and Construction Management, Khulna University of Engineering and Technology, Khulna 9203, Bangladesh; habib@becm.kuet.ac.bd (M.H.R.S.); merazkuet36@gmail.com (M.M.M.)
3 Department of Civil and Architectural Engineering, Aarhus University, 8000 Aarhus, Denmark
* Correspondence: miahmj@cae.au.dk; Tel.: +45-9166-1896

**Abstract:** The utilization of waste products is becoming a vital aspect of the construction industry to safeguard environmental assets and mitigate pollution, all of which lead to long-term sustainable development. From this perspective, this experimental investigation was carried out to determine the cumulative influence of waste glass cullet and metakaolin (MK) as partial replacements for coarse aggregates and cement in an isolated and combined manner. This research demonstrated the influence of integrating glass aggregate and metakaolin wherein coarse aggregate was substituted by 10%, 15%, 20%, 25%, and 30% glass cullet (by weight), and cement was supplemented with 10% metakaolin. The substitution of waste glass with coarse aggregate significantly declines the compressive strength correspondingly; however, the integration of 10% metakaolin powder enhanced the strength slightly for all specimens up to 25%. On the other hand, for flexural strength, the inclusion of glass waste in concrete reduced the performance, whereas the incorporation of metakaolin boosted the strength but did not achieve greater strength compared to the control mixture. The sustainability analysis revealed that the production cost and eCO$_2$ emission could be reduced by 15% and 7% by incorporating glass cullet and metakaolin in the concrete mix, which satisfied sustainability. Based on the experimental results, the ideal proportion substitution would be 25% glass aggregate with 10% metakaolin, which could satisfactorily be used to generate sustainable concrete.

**Keywords:** recycling; glass waste; metakaolin; mechanical characteristics; cost; eCO$_2$ emissions





## 1. Introduction

The concrete industry is questing after the development of alternative materials that mitigate environmental challenges caused by the continual use of environmental assets and emission of greenhouse gas as well as preserve quality, leading to a sustainable environment. Environmental resources utilized as concrete components are scarce in quantity, and concrete might have to be abandoned as a building material at a certain stage in the future if it is utilized at the present rate in the construction sector, resulting in catastrophe and turmoil. The manufacturing of concrete is anticipated to need 9 billion tons of aggregate, 1.5 billion tons of cement, and 1 billion tons of water each year [1]. Because of environmental degradation, carbon emissions, and heavy energy usage, it has a range of detrimental environmental effects, including climate change, scarcity of natural resources, and air pollution [2]. Aggregate is one of the most important elements of concrete, contributing approximately 60–75% of the overall volume of the finished product [3]. The worldwide demand for concrete aggregates is beyond 26.8 billion tons yearly [4].

Consequently, traditional aggregate resources are deteriorating at an accelerated rate to meet this ever-increasing demand for concrete aggregates. Additionally, the production of the primary binding agent, Portland cement, is a high-cost and power-consuming procedure that emits $CO_2$ at a significant amount. For manufacturing one ton of cement, it takes about 1.5 tons of raw elements [5] and 4000 MJ of power [6]. Because of the substantial energy required for generating Portland cement, which produces 5–7% of overall carbon emissions, the impact of employing Portland cement on concrete creation would be tremendous [7]. As a result, integrating waste products as concrete elements would be a crucial choice for making optimal use of environmental assets and minimizing contamination.

Concrete utilization is anticipated to grow to almost 18 billion tons by 2050 as a consequence of enhanced infrastructure development; thus, it is safe to conclude that concrete will play an important part in shaping the future [8]. Moreover, waste formation is followed by a dumping challenge as well as severe environmental repercussions. Additionally, the majority of the waste products are useless. If these products can be integrated as a component in concrete after being evaluated for functionality, it would be tremendously advantageous for developing a sustainable construction industry, employing unproductive waste products, and creating more cost-effective concrete. Numerous studies have been done recently for integrating different waste products, such as demolition wastes [9–11], municipal solid wastes [12–14], and other commercial and industrial wastes, including crushed brick [15–18], plastic [19–21], steel scrap metal [22,23], crushed glass powder [24–26], and electronic waste [27–29] to strengthen concrete in ways that make it safer and capable of performing more various functions. Biological waste (i.e., solidified plant) [30] and polymeric waste (i.e., polystyrene foam) [31] were also subjected to recent studies, resulting in positive outcomes and indicating that the search for novel materials can provide alternatives to conventional concrete materials.

Glass products are some of the most commonly employed products in daily life, and their use is expanding day by day. This increased glass manufacturing and utilization in recent times is attributable to increased urbanization and rapid economic development [32]. Glass is a transparent substance that is generated by liquefying the silica, soda ash, and $CaCO_3$ mixture at elevated temperatures, then chilling the mix until it solidifies without crystallizing [33]. As per UN Projections, the world's solid trash totaled 200 million tons in 2004, with waste glass accounting for 7% of that amount, or 14 million tons [34]. The rising quantities of waste glass, particularly window glass or sheet, have become a significant issue that needs efficient and equitable alternatives. These waste glasses may be repurposed for generating new glass items; however, the sector is confronted with problems leading to more expensive glass reprocessing [35]. As a result, a tiny proportion is repurposed, and the remaining is discarded as waste. Dumping discarded glasses is problematic since they are not compostable [33], rendering these ecologically unfriendly and polluting. As a consequence, its application in other fields is becoming progressively crucial.

While considering the features of waste glass, including absorption and durability, it appears that this could be used as a supplemental binding component or as an aggregate in typical concrete. Glass has a very low absorbency, rendering it a particularly durable substance. Furthermore, glass has such a high stiffness, giving the concrete a strong resistance to abrasion [33]. Numerous studies have looked into the different attributes of concrete with the integration of glass as a substitute for cement [36–38], coarse aggregate [33,39,40], and fine aggregate [41,42] at varying concentrations. The qualities seem to be positive and favorable overall. As per Çelik et al. [43], the concrete mixture, including waste glass powder as fine aggregate, resulted in the proper pozzolanic reaction. The released silica, from glass dissolution, reacts with calcium hydroxide (CH) to form C-(N)-S-H (alkali-silica gel) with different compositions depending on the system. This pozzolanic reaction of glass not only consumes portlandite to form in situ C–S–H, which appears as a reaction rim around glass grains, and precipitated C-S-H but also reduces monosulfate level, which in turn, augmented in enhanced mechanical properties.

Metakaolin (MK) is a residue that is abundantly generated and has high reactivity and silica–aluminous constitution [44]. Kaolin, a raw resource in numerous industries, is transformed into metakaolin using thermal processing that chemically removes bonded water and changes the crystalline phase [45,46]. The thermal system causes moisture to leak out of the kaolin ($Al_2O_3 \cdot 2SiO_2 \cdot 2H_2O$) [47] and changes the structural parameters, leading to metakaolin, an unstructured aluminosilicate ($Al_2O_3 \cdot 2SiO_2$) [48]. At room temperature, metakaolin combines with $Ca(OH)_2$ to generate the Calcium Silicate Hydrate (C-S-H) gel, which then interacts with Calcium Hydroxide (CH) to create alumina-containing stages, such as $C_4AH_{13}$, $C_2ASH_8$, and $C_3AH_6$ [49,50]. Since MK is a pozzolanic substance, it lowers the amount of CH inside a cement-based process. In contrast, cement is substituted by MK, the blend of pore improvement and decrease in CH, resulting in enhanced durability [51,52]. This has the potential to be both cost-effective and environmentally friendly.

Metakaolin's unique chemical composition allows it to function as a perfect additional cementing component among various nutrients and residues already used in construction such as various powder contents [53]. While employed as a partial substitution for cement, it gives advantages in terms of mechanical strength and tolerance to the intrusion of hostile substances [54]. Tafraoui et al. [55] discovered that the improvements in mechanical properties and durability of concrete made with MK are comparable to the advances made with silica fume. Additionally, Duan et al. [56] compared the compressive strength enhancements in concrete made employing MK, silica fume, and pulverized granulated blast-furnace slag and discovered that concrete made with MK seemed to have the best results. Siddique and Kaur [57] investigated the impact of MK as incorporation of cement at weight ratios of 0%, 5%, 10%, and 15%. The compressive strength reduced at 15% MK replacement, whereas 10% MK content was shown to be the best replacement amount. Moreover, Dinakar et al. [58] studied the mechanical and endurance features of concrete integrating 5%, 10%, and 15% MK, concluding that 10% was the optimum amount with respect to the compressive strength and that the durability increases proportionately with the enhancement of the proportion.

*Research Significance*

Based on the increasing deterioration of the present and emerging ecosystem, academics and engineers have been interested in the long-term sustainable development of concrete. Waste glass aggregate and MK have been employed in the development of self-compacting concrete in an isolated manner and revealed positive outcomes. Previously, numerous studies were conducted on improving the performance of concrete made with fine glass aggregate and dust. However, only a few investigations have been done on the utilization of waste glass cullet as a coarse aggregate. In addition, employing glass aggregate from domestically supplied waste glass bottles as coarse aggregate along with metakaolin has not been initiated yet. Hence, the objective of this analysis is to identify the behavior, features, and efficiency of concrete with waste glass as supplementary coarse aggregate and metakaolin as supplementary cementitious material. This study also focuses on promoting the minimization of environmental problems associated with employing natural resources by suggesting substituting them with less environmentally damaging elements that can prolong the structure's life span instead of decreasing productivity and minimizing emissions.

## 2. Materials and Methods

### 2.1. Research Plan

Experimental investigations of the present study include concrete's fresh qualities, mechanical characteristics, carbon analysis, and cost estimation, which are all evaluated and reported. Figure 1 illustrates the successive activities through which these investigations were carried out.

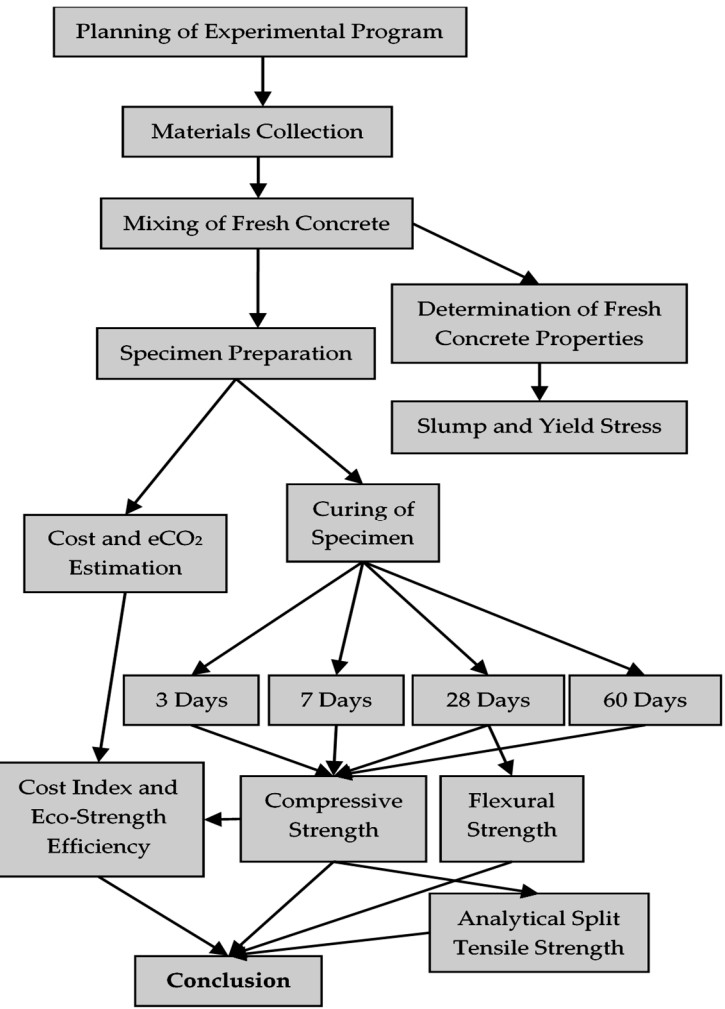

**Figure 1.** Research methodology flow chart.

## 2.2. Materials

To conduct the present research, the Type I ordinary Portland Cement (OPC) of the "Seladang" brand, manufactured by Tenggara Cement Manufacturing Sdn, was used as a primary binder for the concrete mixes that correspond to the ASTM C150/C150M-20 [59] standard. In addition, 10% metakaolin was employed as a supplementary cementing agent because prior literature [57,58,60] had revealed that incorporating 10% metakaolin as a replacement for cement produces the best performance. To manufacture metakaolin, the yellowish-white-tinted regional kaolin provided by "Kaolin Malaysia Sdn. Bhd." was heat-treated at 750 °C in a dedicated oven. The chemical composition of cement and kaolin used in the investigation is given in Table 1.

The research utilized river sand as fine aggregate, which was acquired from a regional supermarket and air-dried to achieve a saturated surface dry condition for maintaining the water–cement ratio, and granite crushed aggregates as coarse aggregate (CA), which were extracted from local producers with a marginal diameter of 10 mm and met the requirements of ASTM C136/C136M-19 [61]. Glass material (cullet) utilized as additional coarse aggregate was gathered in various forms, dimensions, patterns, and colors from houses, clubs, and bars, and municipal trash collection authorities because it is not easily accessible in a consistent queue regionally. The material was collected, rinsed to remove any sticky material that had accumulated on the glass, and then allowed to dry completely. The waste glass was then crushed and sieved to achieve a grade that was close to that of conventional coarse aggregate, as defined by ASTM C136/C136M-19 [61]. Figure 2 illustrates a sample of waste glass cullet that was employed as supplemental coarse aggregate in this study at

various amounts. The gradation curve of stone chips as coarse aggregate, crushed glass aggregate, and fine aggregate (sand), is depicted in Figure 3 with upper and lower limits in accordance with ASTM-C33 [62].

**Table 1.** The chemical composition of cement and kaolin.

| Parameters | Cement (%) | Kaolin (%) |
|---|---|---|
| Silica ($SiO_2$) | 20.1 | 55 |
| Alumina ($Al_2O_3$) | 4.9 | 29 |
| Iron Oxide ($Fe_2O_3$) | 2.5 | 1 |
| Calcium Oxide (CaO) | 65 | - |
| Magnesium Oxide (MgO) | 3.1 | 0.5 |
| Sodium ($Na_2O$) | 0.2 | 0.02 |
| Potassium Oxide ($K_2O$) | 0.4 | 3.1 |
| Titanium Dioxide ($TiO_2$) | 0.2 | - |
| Loss On Ignition | 2.5 | 8.8 |
| Fineness (2 μm) | 2.4 | 10 |

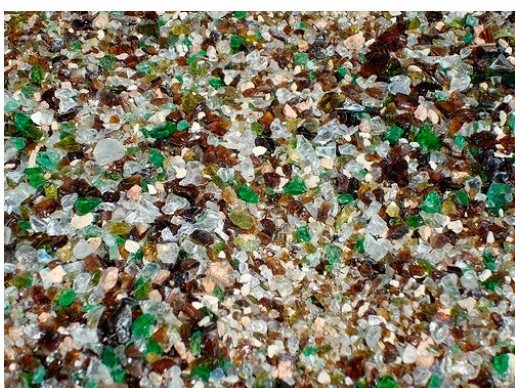

**Figure 2.** Sample of waste glass cullet used in the investigation.

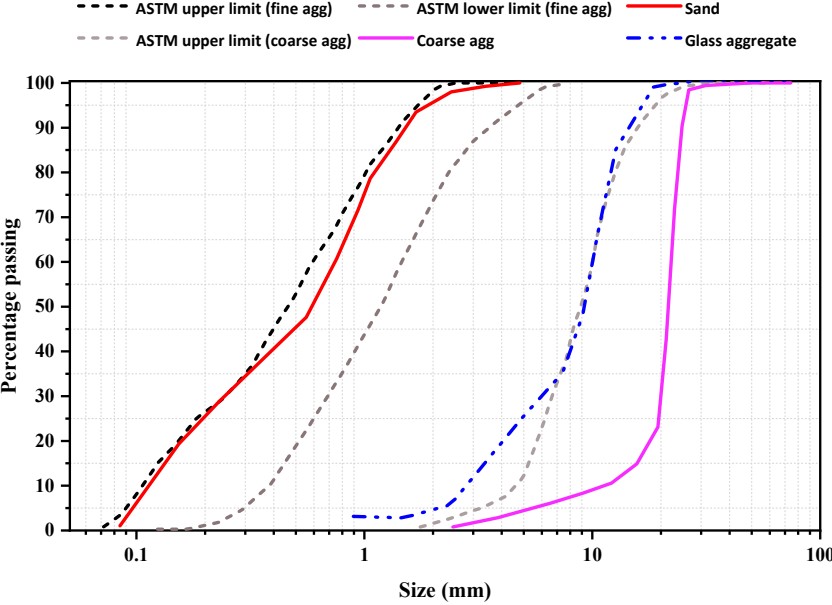

**Figure 3.** Gradation Curve of the aggregates.

*2.3. Mix Proportioning and Specimen Preparation*

A total of six combinations for both with and without metakaolin were constructed for this research in addition to the control sample to evaluate the impacts of the waste

glass cullet and MK and compare the qualities with the control specimen. In this study, the coarse aggregate and cement were substituted with glass aggregate (10%, 15%, 20%, 25%, and 30% by weight of coarse stone aggregate) and 10% metakaolin. The water–cement ratio in such concrete mixtures was set at 0.45, and the compressive strength target was set at 30 MPa after 28 days. The control sample, designated as GC 0 and GMC 0 for without and with metakaolin, respectively, was the first sample of concrete utilized for evaluating and analyzing concrete incorporating varied proportions of cullet. For this experiment, the remaining batches were molded integrating 10%, 15%, 20%, 25%, and 30% waste glass aggregates and labeled as GC 10, GC 15, GC 20, GC25, and GC 30 for combinations without metakaolin. In contrast, GMC 10, GMC 15, GMC 20, GMC 25, and GMC 30 for combinations with 10% metakaolin. The concrete mixtures with various waste glass aggregate percentages with and without metakaolin are demonstrated in Table 2.

**Table 2.** Ingredient proportions of the concrete mixtures (kg/m$^3$).

| Mix Id | Cullet Percentage | Water (Kg/m$^3$) | Cement (Kg/m$^3$) | Metakaolin (Kg/m$^3$) | Fine Aggregate (Kg/m$^3$) | Coarse Aggregate (Kg/m$^3$) | Cullet (Kg/m$^3$) |
|---|---|---|---|---|---|---|---|
| GC 0 | Control | 177.75 | 395 | - | 805 | 989 | 0 |
| GC 10 | 10% | 177.75 | 395 | - | 805 | 890.1 | 98.9 |
| GC 15 | 15% | 177.75 | 395 | - | 805 | 840.65 | 148.35 |
| GC 20 | 20% | 177.75 | 395 | - | 805 | 791.2 | 197.8 |
| GC 25 | 25% | 177.75 | 395 | - | 805 | 741.75 | 247.25 |
| GC 30 | 30% | 177.75 | 395 | - | 805 | 692.3 | 296.7 |
| GMC 0 | 0% | 177.75 | 355.5 | 39.5 | 805 | 989 | 0 |
| GMC 10 | 10% | 177.75 | 355.5 | 39.5 | 805 | 890.1 | 98.9 |
| GMC 15 | 15% | 177.75 | 355.5 | 39.5 | 805 | 840.65 | 184.35 |
| GMC 20 | 20% | 177.75 | 355.5 | 39.5 | 805 | 791.2 | 197.8 |
| GMC 25 | 25% | 177.75 | 355.5 | 39.5 | 805 | 741.75 | 247.25 |
| GMC 30 | 30% | 177.75 | 355.5 | 39.5 | 805 | 692.3 | 296.7 |

GC = Glass replacement; GMC = Glass replacement with 10% metakaolin.

For the fabrication of concrete, the combination, the binders (cement and metakaolin) are initially measured and physically combined until the elements were homogeneously blended together to ensure the mixture's consistency. The binders and aggregates are then put into the mixing machine, pouring a small volume of liquid; the mixer is started for the first 20 s. Then, the remaining water is added, and the mixture is allowed to mix for 2 min before being tested. Compressive strength was measured using 100 mm × 200 mm cylinder specimens, whereas flexural strength was examined using 100 mm × 100 mm × 500 mm prisms that were prepared in the laboratory. The molds were extracted after 24 h of casting, and the samples were placed in the water tank to cure at 27 ± 2 °C.

### 2.4. Testing Methods

The Slump test was used to examine the consistency of the concrete and evaluate its workability by comparing the control sample and other combinations, according to the ASTM C143/C143M-20 [63] specification. The compressive strength was assessed at 3, 7, 28, and 60 days using a 100 mm × 200 mm cylinder in accordance with ASTM C39/C39M-20 [64]. Additionally, the flexural strength was evaluated using a prism sizing 100 mm × 100 mm × 500 mm employing third-point flexural loading at 7, 28, and 60 days following ASTM C78/C78M-18 [65] standard.

### 3. Results and Discussion

#### 3.1. Rheological Properties

3.1.1. Slump and Yield Stress

Figure 4 provides a graphical illustration of the slump and yield stress values of all concrete mixes containing different percentages of glass aggregate and a constant percentage of MK. According to the results, an increasing tendency for the slump value was

observed with the increased percentages of glass aggregate. The maximum values were recorded for the mix fabricated with 30% glass aggregate (i.e., GC 30 and GMC 30). The control concrete mix with 0% glass aggregate (GC 0) exhibited a minimum slump value of 39 mm, and the control concrete mix for glass metakaolin concrete with 0% glass aggregates (GMC 0) showed a slump value of 40 mm, which is quite same to the former one. This increasing pattern of workability could be due to the smoother surface of glass coarse aggregates than crushed stone aggregate, which forms a poorer cohesion with cement paste. These properties can dramatically lessen the inter-particle resistance among the glass aggregates and other constituents of concrete in the mix, thus achieving a smooth flow caused by minor needed energy to overcome frictional stress in the freshly mixed concrete, leading to a significantly higher slump to the mix. This behavior is more pronounced with the rising percentages of glass aggregate in the concrete mixes. Generally, glass particle has a lower absorption capacity, which can leave the free water in the mix and assist in improving the inter-particle movements in the matrix and improve the slump induced by the more promising ball-bearing effect. This increasing pattern of workability was also observed in the studies of Tian et al. [66] and Gerges et al. [67].

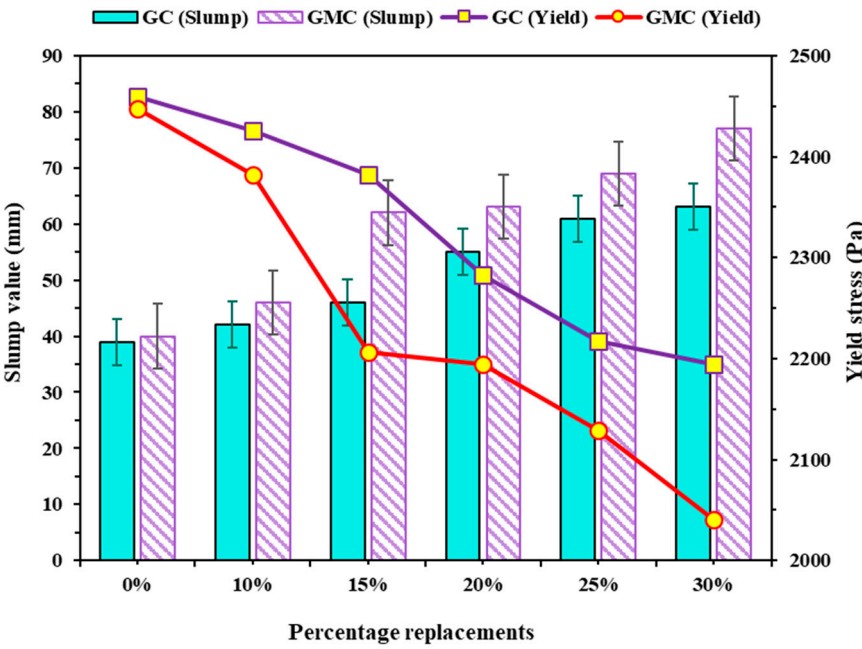

**Figure 4.** Slump and yield stress values for all concrete mixes.

It was registered that the concrete containing MK augmented the concrete slump more than the MF-free concrete mixes, as reported in Figure 4. This could be due to the slower and lower magnitude of the temperature inside the freshly mixed concrete containing MK than the concrete cast with OPC. Indeed, a faster development with higher temperature could be expected for the OPC concrete due to the rapid and higher heat of hydration of OPC, thus lowering the workability of concrete mixes. On the contrary, the lower temperature in the MK concrete mixes could decelerate the cement paste's hydration and enhance the slump. Certainly, the higher workability of concrete mixes could dramatically improve the transportability and moveability; thus, they can help to compact the concrete better and reduce the porosity due to the formation of lower voids and permeability, enhancing the global strength and durability properties of concrete.

The yield stress for all the mixes was achieved without performing any experimental investigation, as there is a procedure suggested by Tattersall and Banfill [68]. The numerical data for yield stress adopted was characterized by two equations (Equations (1) and (2)) both for GC and GMC concrete which shows a 99% correlative confidence. The relationship

between the slump and yield stress for both GC and GMC is presented in Figure 5 and denoted by Equations (1) and (2), respectively.

$$f_y = -11.49\,S + 2888 \tag{1}$$

$$f_y = -11.49\,S + 2888 \tag{2}$$

where, $f_y$ = yield stress (pa), and $S$ = slump value (mm)

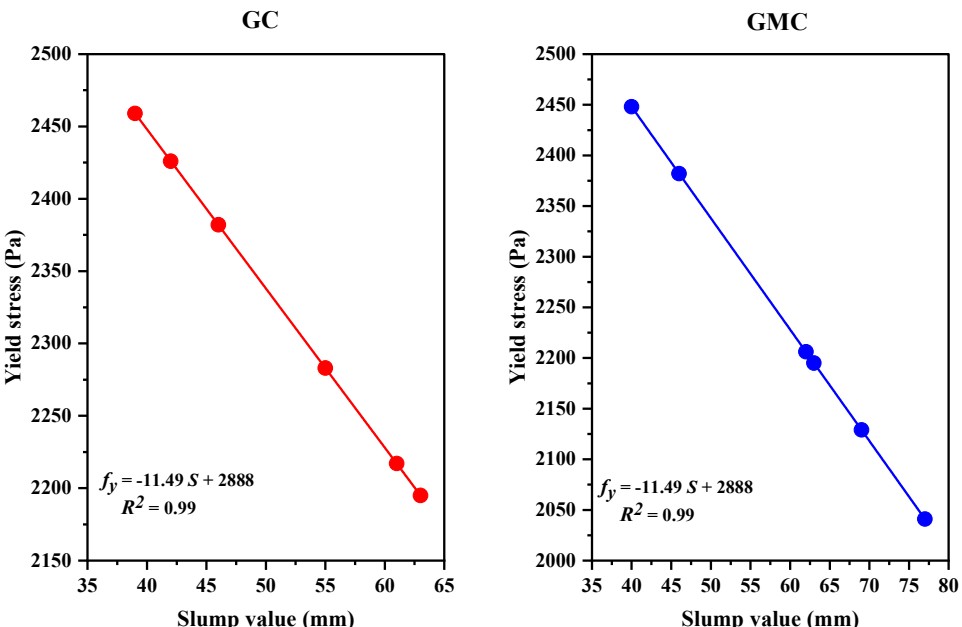

**Figure 5.** Yield stress of concrete mixture in relation to the slump value.

A linear relationship was registered and revealed that the slump values decreased with the increased value of yield stress of the freshly mixed concrete.

### 3.2. Mechanical Properties

### 3.2.1. Effect of Crushed Glass Aggregates on Compressive Strength for GC

Table 3 and Figure 6 represent the compressive strength of GC with various proportions of crushed glass aggregates tested at 3, 7, 28, and 60 days. Numerous statistical parameters were also calculated from the attained results, such as coefficient of variation (CoV), standard deviation, standard error, and 95% confidence interval within 3, 7, 28, and 60 days. It is observable from the table that the mean compressive strength varied from 16.99 to 20.97 MPa for 3 days and 19.23 to 24.47 MPa; 24.28 to 30.35 MPa; 28.87 to 33.3 MPa for the 7th, 28th, and 60th days, respectively. The standard deviation for GC concrete mixes varied from 0.74 to 1.48, along with a coefficient of variation spanning from 0.017% to 0.059% and a standard error of 0.26 to 0.86. At 3 days, GC10 achieved the minimum compressive strength of 16.99 MPa, including a 95% confidence interval of 14.69 MPa to 19.29 MPa. In contrast, GC25 showed a maximum compressive strength of 20.97 MPa, including a 95% confidence interval of 19.14 MPa to 22.80 MPa. Similar results were found at 7 days of curing. However, for 28 days, GC 10 attained the maximum compressive strength of 30.35 MPa. On the contrary, GC 30 exhibited a minimum compressive strength of 24.28 MPa, and for 60 days, GC 10 displayed a maximum compressive strength of 33.30 MPa, as shown in Table 3 and Figure 6.

**Table 3.** Summary of the compressive strength test results for GC.

| Mix Id | Curing Days | Mean Strength (MPa) | Standard Deviation | CoV (%) | Standard Error | 95% Confidence Interval | |
|---|---|---|---|---|---|---|---|
| | | | | | | Lower Range | Upper Range |
| GC 0 | 3 | 17.97 | 0.87 | 0.048 | 0.50 | 15.81 | 20.13 |
| | 7 | 19.93 | 1.11 | 0.056 | 0.64 | 17.17 | 22.69 |
| | 28 | 28.91 | 1.05 | 0.036 | 0.61 | 26.30 | 31.52 |
| | 60 | 31.93 | 1.23 | 0.039 | 0.71 | 28.87 | 34.99 |
| GC 10 | 3 | 16.99 | 0.92 | 0.054 | 0.53 | 14.69 | 19.29 |
| | 7 | 19.23 | 1.06 | 0.055 | 0.61 | 16.59 | 21.87 |
| | 28 | 30.35 | 1.16 | 0.038 | 0.67 | 27.47 | 33.23 |
| | 60 | 33.30 | 1.24 | 0.037 | 0.72 | 30.22 | 36.38 |
| GC 15 | 3 | 17.38 | 0.99 | 0.057 | 0.57 | 14.93 | 19.83 |
| | 7 | 20.59 | 1.04 | 0.051 | 0.60 | 18.00 | 23.18 |
| | 28 | 29.90 | 1.18 | 0.039 | 0.68 | 26.97 | 32.83 |
| | 60 | 32.54 | 1.48 | 0.046 | 0.86 | 28.85 | 36.23 |
| GC 20 | 3 | 18.82 | 0.77 | 0.041 | 0.44 | 16.91 | 20.73 |
| | 7 | 21.31 | 0.98 | 0.046 | 0.57 | 18.87 | 23.75 |
| | 28 | 25.73 | 0.45 | 0.017 | 0.26 | 24.62 | 26.84 |
| | 60 | 30.06 | 1.03 | 0.034 | 0.60 | 27.50 | 32.62 |
| GC 25 | 3 | 20.97 | 0.74 | 0.035 | 0.43 | 19.14 | 22.80 |
| | 7 | 24.47 | 1.33 | 0.054 | 0.77 | 21.17 | 27.77 |
| | 28 | 26.38 | 1.15 | 0.044 | 0.67 | 23.51 | 29.25 |
| | 60 | 31.66 | 1.24 | 0.039 | 0.71 | 28.59 | 34.73 |
| GC 30 | 3 | 20.33 | 1.05 | 0.051 | 0.60 | 17.73 | 22.93 |
| | 7 | 22.99 | 1.36 | 0.059 | 0.79 | 19.61 | 26.37 |
| | 28 | 24.28 | 1.02 | 0.042 | 0.59 | 21.75 | 26.81 |
| | 60 | 28.87 | 1.41 | 0.049 | 0.81 | 25.38 | 32.36 |

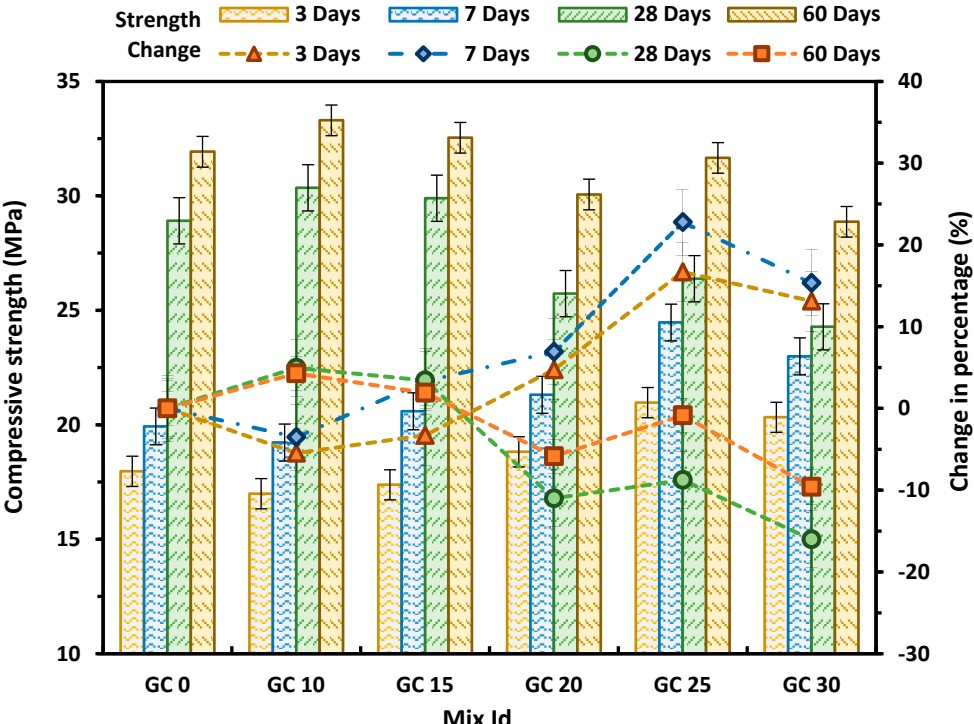

**Figure 6.** Compressive strength of concrete mixes for GC at 3, 7, 28, and 60 days.

Almost 17% and 23% increments of compressive strength were observed for GC 25 for 3 and 7 days, respectively; however, the strength decreased at 28 and 60 days. GC 10 and GC 15 provided the increment of compressive strength for 28 and 60 days at a range of 2 to 5%. The maximum compressive strength was observed for GC 10 for 28 days which was about 5% higher than the control mix. In addition, more than a 4% increment of compressive strength was observed for GC 10 at 60 days. The results reveal that the incorporation of the lower content of glass aggregate provides a positive impact on the compressive strength of concrete. In comparison, the strength decreased for the higher content of glass aggregate in the concrete. Notably, the compressive strength for GC 25 was higher than GC 20 but lower than the control mix. This increase can be attributed to the suitable compaction of the glass aggregate. Moreover, it is possible that the angular-shaped aggregates were well distributed and oriented along with the natural stone aggregate. The combined effects can impart a stiffer matrix to the concrete, resulting in higher strength for 25% GC incorporation. Similar irregularities in compressive strength of higher contents of waste glass were observed by Arabi et al. [69], when they used windshield waste glass aggregate to replace natural coarse aggregate partially. In their study, a notable jump in strength was seen when the glass aggregate was used at a higher dosage along with recycled natural coarse aggregate. The enhancement in compressive strength could be due to the high Mohs hardness index of glass particles. In contrast, generally, glass aggregates are fairly flat and flaky, which could provide inhomogeneity in their distribution in the concrete mix, resulting in higher porosity and permeability. Indeed, the flat and flaky shape of glass aggregates is more vulnerable to break/damage (i.e., easy to break with lower load due to the higher slenderness effect) when they experience the mechanical load, which also depends on their orientation [70]. Additionally, the glass aggregate has a smooth surface, which can create a poor interfacial transition zone (ITZ) around the glass aggregates and cement paste [71]. Last but not least, glass aggregate has significantly high brittleness, a great disadvantage of glass particles. Hence, when the concrete experiences the mechanical load, the cracks can easily initiate at the ITZ, and then the cracks propagate more effortlessly as the glass aggregates are more vulnerable to detaching from the cement mortar, i.e., failure may occur primarily due to the initiation of cracks at ITZ rather than the glass and stone aggregate fracture [72]. This behavior prevents the concrete from resisting the higher mechanical load and presents a lower compressive strength to the concrete cast with glass aggregates. As a result, 10% replacement of glass aggregate displayed the optimum result. A similar compressive strength pattern can also be found in the study carried out by Gerges, Issa, Fawaz, Jabbour, Jreige and Yacoub [67]. Ganiron Jr. [73] suggests a 5% replacement of glass aggregates for enhancing compressive strength; in addition, a maximum of up to 20% replacement is suggested by Tian, Liu, Cui, Sun, Wang, Li, Fu and Wang [66].

### 3.2.2. Effect of Crushed Glass Aggregates on Compressive Strength for GMC

Table 4 represents the compressive strength of GMC with various proportions of crushed glass aggregates. Meanwhile, Figure 7 illustrates the compressive strength and change in the strength of concrete mixed fabricated with different percentages of glass aggregates and MK tested at 3, 7, 28, and 60 days. Numerous statistical parameters were also calculated from the attained results, such as coefficient of variation (CoV), standard deviation, standard error, and 95% confidence interval within 3, 7, 28, and 60 days, respectively. It was found that the mean compressive strength varied from 15.28 to 22.54 MPa, for 3 days and 20.5 to 29.61 MPa; 27.64 to 38.39 MPa; 32.34 to 39.73 MPa for the 7th, 28th, and 60th days, respectively. The standard deviation for GMC concrete mixes varied from 0.67 to 1.34, along with a coefficient of variation spanning from 0.018% to 0.058% and a standard error of 0.39 to 0.77. At 3 days, GMC30 accomplished the minimum compressive strength of 15.28 MPa, including a 95% confidence interval of 13.08 MPa to 17.48 MPa, whereas GMC15 achieved the maximum compressive strength of 22.54 MPa, including a 95% confidence interval of 20.01 MPa to 25.07 MPa. Similar results were found at 7 days of curing. However,

for 28 days, GMC25 attained the maximum compressive strength of 38.39 MPa, whereas GMC30 exhibited the minimum compressive strength of 27.64 MPa, and for 60 days where the GMC25 displayed the maximum compressive strength of 39.73 MPa.

**Table 4.** Summary of the compressive strength test results for GMC.

| Mix Id | Curing Days | Mean Strength (MPa) | Standard Deviation | CoV (%) | Standard Error | 95% Confidence Interval | |
|---|---|---|---|---|---|---|---|
| | | | | | | Lower Range | Upper Range |
| GMC 0 | 3 | 19.57 | 0.87 | 0.044 | 0.50 | 17.41 | 21.73 |
| | 7 | 20.50 | 0.68 | 0.033 | 0.39 | 18.82 | 22.18 |
| | 28 | 29.70 | 0.75 | 0.023 | 0.43 | 30.63 | 34.37 |
| | 60 | 32.45 | 0.82 | 0.024 | 0.47 | 32.23 | 36.29 |
| GMC 10 | 3 | 20.18 | 0.67 | 0.033 | 0.39 | 18.51 | 21.85 |
| | 7 | 24.60 | 0.91 | 0.037 | 0.52 | 22.35 | 26.85 |
| | 28 | 29.80 | 1.23 | 0.041 | 0.71 | 26.75 | 32.85 |
| | 60 | 32.67 | 1.13 | 0.035 | 0.65 | 29.85 | 35.49 |
| GMC 15 | 3 | 22.54 | 1.02 | 0.045 | 0.59 | 20.01 | 25.07 |
| | 7 | 29.61 | 0.96 | 0.032 | 0.55 | 27.23 | 31.99 |
| | 28 | 31.31 | 1.06 | 0.034 | 0.61 | 28.68 | 33.94 |
| | 60 | 33.55 | 1.34 | 0.040 | 0.77 | 30.23 | 36.87 |
| GMC 20 | 3 | 22.12 | 0.89 | 0.040 | 0.51 | 19.92 | 24.32 |
| | 7 | 27.22 | 0.98 | 0.036 | 0.56 | 24.79 | 29.65 |
| | 28 | 35.77 | 1.19 | 0.033 | 0.69 | 32.81 | 38.73 |
| | 60 | 37.21 | 1.14 | 0.031 | 0.66 | 34.37 | 40.05 |
| GMC 25 | 3 | 19.86 | 0.84 | 0.042 | 0.49 | 17.77 | 21.95 |
| | 7 | 24.81 | 0.92 | 0.037 | 0.53 | 22.51 | 27.11 |
| | 28 | 38.39 | 0.69 | 0.018 | 0.40 | 36.67 | 40.11 |
| | 60 | 39.73 | 0.96 | 0.024 | 0.55 | 37.36 | 42.10 |
| GMC 30 | 3 | 15.28 | 0.89 | 0.058 | 0.51 | 13.08 | 17.48 |
| | 7 | 22.06 | 0.95 | 0.043 | 0.55 | 19.69 | 24.43 |
| | 28 | 27.64 | 1.31 | 0.047 | 0.75 | 24.39 | 30.89 |
| | 60 | 32.34 | 1.09 | 0.034 | 0.63 | 29.63 | 35.05 |

After 3 and 7 days of the curing period, GMC 15 achieved the maximum compressive strength of 22.54 MPa and 29.61 MPa, respectively. However, after 28 and 60 days, GMC 25 achieved the maximum compressive strength of 38.39 MPa and 39.71 MPa, respectively. The addition of metakaolin enhances the strength of concrete mixes, i.e., a gradual increment of compressive strength increment was observed for 10%, 15%, 20%, and 25% glass aggregate incorporation. GMC 25 exhibited the highest increment of compressive strength compared to the control mix at 28 days. Almost 30% and 23% increments of compressive strength were observed for GMC 25 after the curing of 28 and 60 days, respectively. However, a decremental pattern of strength was observed for GMC 30 compared to the control mix. A constant 10% replacement of metakaolin helps to initiate the filler effect as well as the compounding effect of concrete mix, which was observed by Sujjavanich et al. [74]. In the study of Ramezanianpour and Bahrami Jovein [75], the early reaction property of metakaolin was observed to form a C-S-H bond earlier, and the addition of 10–15% of metakaolin exhibited an optimum result. With the effect of metakaolin, a 25% replacement of glass aggregate displayed the highest compressive strength. The strength increment pattern was also observed in the study of Afshinnia and Rangaraju [76] and Lee et al. [77], and suggested a 15–25% of glass aggregate replacement to enhance the strength.

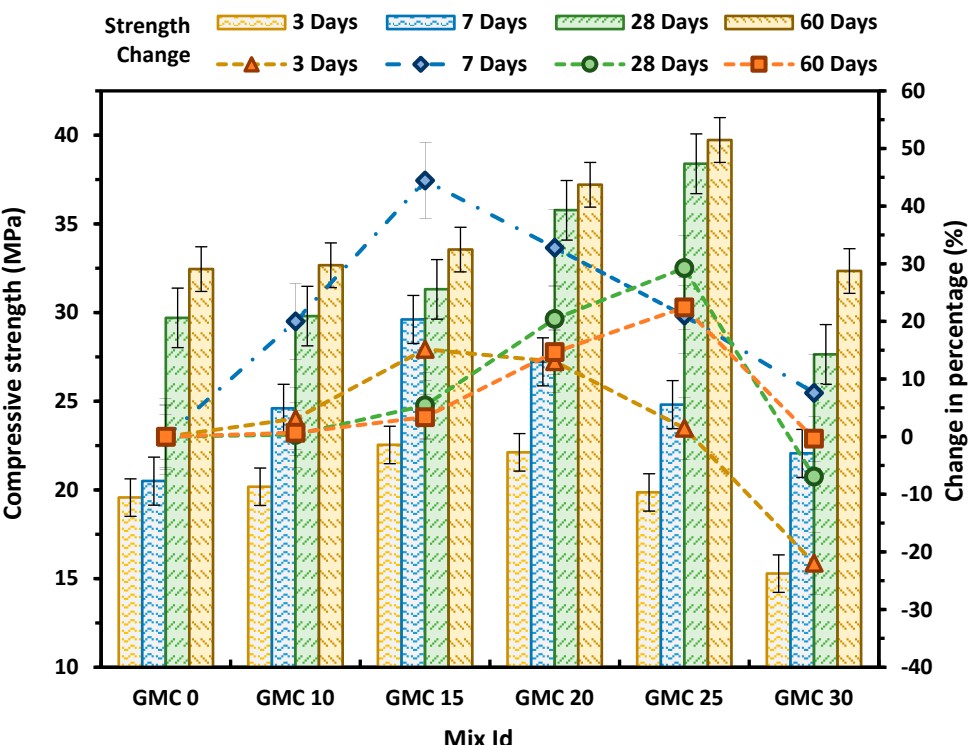

**Figure 7.** Compressive strength of concrete mixes for GMC at 3, 7, 28, and 60 days.

The enhancement in compressive strength of concrete cast with 10% MK as replacement of cement and glass aggregate is associated with the combined refinement of pore structure due to the micro-filler effect (i.e., lowering porosity, resulting in an enhanced ITZ) and formation of secondary C-S-H gels in the cement matrix due to the pozzolanic reaction of MK with calcium hydroxide. Indeed, in control concrete, the strength development depends on only the hydration reaction of clinker; in contrast, OPC-MK concrete combines OPC hydration and Mk's pozzolanic action. Especially at prolonged curing age (i.e., 60 days), MK can respond with the hydrated products of clinker (CH) with the existence of water in the concrete matrix and could form secondary C-S-H gels, thus dramatically diminishing the size of pores (i.e., decrease the porosity) and connectivity (i.e., lowering the permeability) and boost the ITZ. These properties augment the capability to withstand the higher compressive force and provide greater strength to the concrete containing MK. This behavior does not occur for the concrete cast with 100% OPC (i.e., control concrete), thus making lower compressive strength than the MK-OPC concrete (i.e., GMC). Moreover, the significant jump in the strength of GMC 25 concrete mix can be attributed to the hydration heat of MK based concrete. At early ages, hydration heat can affect the compressive strength of concrete by increasing shrinkage and thermal cracks [78]. These effects can be intensified using MK, a high-reactivity pozzolan [79]. Along with this, as discussed in Section 3.1.1, incorporating glass aggregate increases the workability and makes more water available for the hydration process. As a result, the higher dosage of GC exhibited comparatively lower strengths at the early ages of 3 and 7 days, whereas higher strengths at later ages of 28 and 60 days.

### 3.2.3. Analysis of Variance (ANOVA) for Compressive Strength Test Results

A single-factor ANOVA test was performed to assess the statistical significance of GC and GMC mixes on the compressive strength of concrete. The test was initiated for all the curing intervals to observe their statistical significance at a 95% significance level ($\alpha = 0.05$). The results for both the GC and GMC mix combinations are detailed in Table 5. The null hypothesis states that the percentages of glass cullets in GC and GMC mixes have no statistical significance to their compressive strengths, such as, if the *p*-value is

less than 0.05, it can be rejected, which indicates that a statistically significant relationship exists among the two variables. From the test outcomes, it can be seen that at early curing intervals of 3 and 7 days, there is no statistically significant relationship for both the GC and GMC mixes. In contrast, at early curing days of 28 and 60, a significant statistical relationship can be found. This test outcomes are in line with the findings discussed in Sections 3.2.1 and 3.2.2, where it was evident that the glass cullet and metakaolin both affected the later age compressive strengths more than their earlier ages.

**Table 5.** ANOVA test results for different curing intervals of GC and GMC mixes to their mean compressive strengths.

| Group | Curing Days | Source of Variation | Degree of Freedom | Sum of Squares | Mean Square | F-Test | *p*-Value | Significance |
|---|---|---|---|---|---|---|---|---|
| GC% to Compressive Strength | 3 | Between Groups | 1 | 12.94 | 12.94 | 0.217 | 0.65135 | No |
| | | Within Groups | 10 | 596.35 | 59.63 | | | |
| | 7 | Between Groups | 1 | 67.78 | 67.78 | 1.124 | 0.31391 | No |
| | | Within Groups | 10 | 602.82 | 60.28 | | | |
| | 28 | Between Groups | 1 | 358.07 | 358.07 | 5.833 | 0.03637 | Yes |
| | | Within Groups | 10 | 613.91 | 61.39 | | | |
| | 60 | Between Groups | 1 | 650.62 | 650.62 | 10.902 | 0.00799 | Yes |
| | | Within Groups | 10 | 596.79 | 59.68 | | | |
| GMC% to Compressive Strength | 3 | Between Groups | 1 | 31.85 | 31.85 | 0.516 | 0.48883 | No |
| | | Within Groups | 10 | 616.76 | 61.68 | | | |
| | 7 | Between Groups | 1 | 198.45 | 198.45 | 3.109 | 0.10835 | No |
| | | Within Groups | 10 | 638.36 | 63.84 | | | |
| | 28 | Between Groups | 1 | 758.59 | 758.59 | 11.476 | 0.00691 | Yes |
| | | Within Groups | 10 | 661.02 | 66.10 | | | |
| | 60 | Between Groups | 1 | 1003.94 | 1003.94 | 16.044 | 0.00250 | Yes |
| | | Within Groups | 10 | 625.74 | 62.57 | | | |

### 3.2.4. Comparison of Analytical Values of the Splitting Tensile Strength

To forecast tensile strength relying on compressive strength, various standards prescribe different alternative formulae. The ACI 363R, ACI 318, CEB-FIP, and AS 3600 [80–83] standards were employed to calculate the tensile strength of GC and GMC concrete with respect to the compressive strength of different formulas, as reported by Equations (3)–(6).

$$f_{st} = 0.3 \, (f_c)^{2/3} \text{ (CEB-FIP)} \tag{3}$$

$$f_{st} = 0.59 \sqrt{f_c} \text{ (ACI 363R)} \tag{4}$$

$$f_{st} = 0.56 \sqrt{f_c} \text{ (ACI 318)} \tag{5}$$

$$f_{st} = 0.4 \, \sqrt{f_c} \text{ (AS 3600)} \tag{6}$$

where, $f_{st}$ = Splitting tensile strength (MPa), $f_c$ = Compressive strength (MPa).

Figures 8 and 9 represent the analytical value of splitting tensile strength, which was compared to different standards and plotted against experimental compressive strength. From the analytical result, the ACI 363R recommended standard exhibited the maximum tensile strength value for both the GC and GMC mixes. On the other hand, the AS 3600 exhibited the lowest values for both GC and GMC. These analytical values can be utilized

to measure the splitting tensile strength of GC and GMC, complying with various code standards.

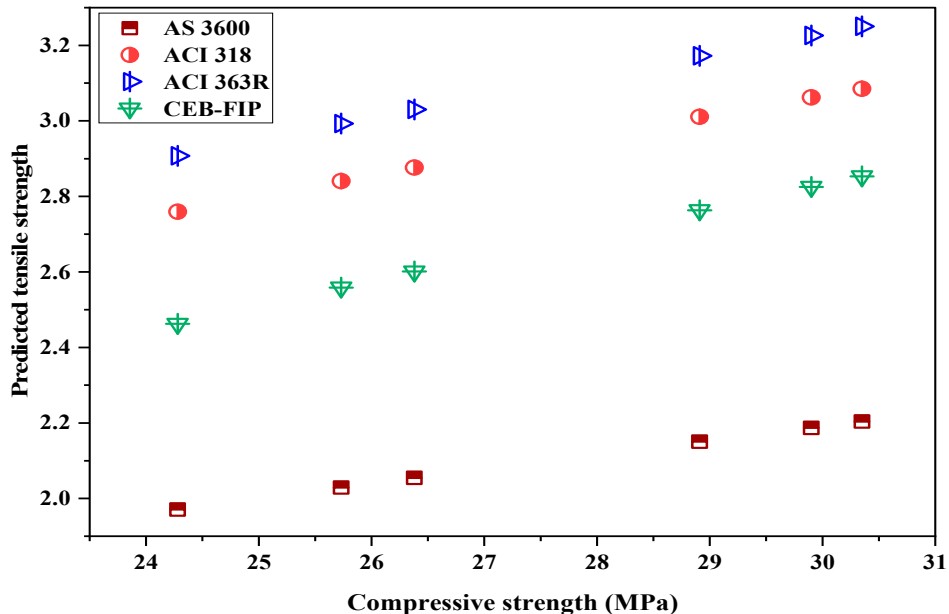

**Figure 8.** Comparison of analytical tensile strength for GC.

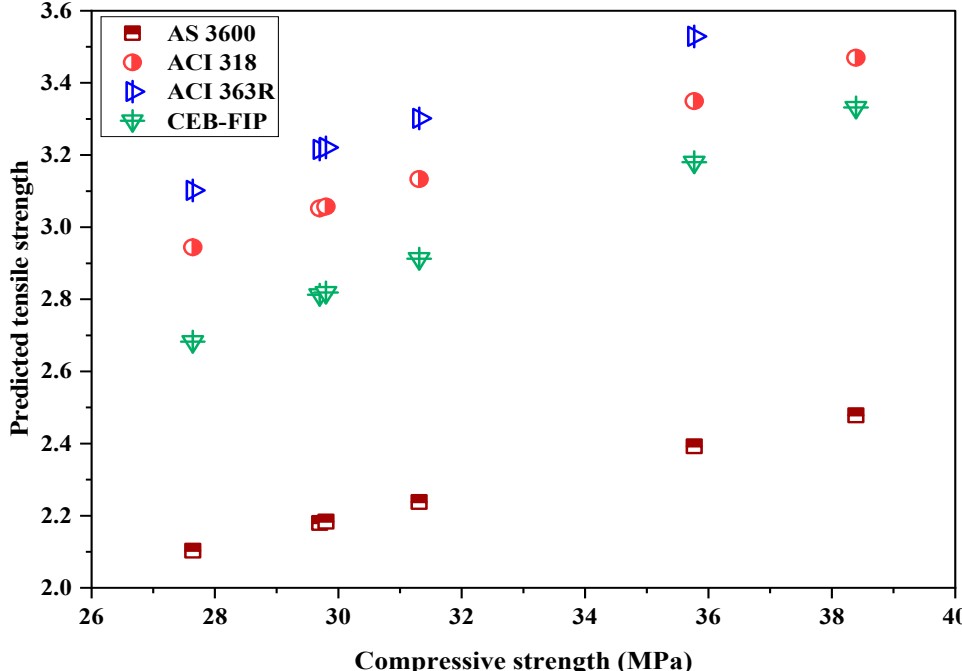

**Figure 9.** Comparison of analytical tensile strength for GMC.

3.2.5. Effect of Metakaolin and Crushed Glass Aggregates on Flexural Strength

Figure 10 graphically demonstrates the mean flexural strength and their fluctuations for GC and GMC at 28 days. Without the mixing of metakaolin, GC 10 achieved the maximum flexural strength of 4.81 MPa, which was 0.63% higher in comparison to the control mix. Above that replacement, the flexural strength decreased with the increased content of glass aggregate in the concrete. As mentioned in Section 3.2.1, the flat and flaky-shaped GC can affect the matrix positively by interlocking and providing load transfer pathways. For this reason, incorporating GC at a lower percentage of 10% exhibited slightly

higher flexural strength. However, when the GC content is further increased, this advantage can turn into a disadvantage because the overuse of GC can impart inhomogeneity in their orientation [70]. In addition, the smooth surface of GC can create poor and porous ITZ between aggregates and cement paste [71]. The combined effect of these can significantly decrease the flexural strength of hardened concrete when GC is employed. However, the addition of metakaolin and glass aggregate decreased the flexural strength of all the GMC mixes with respect to the control mix (GMC 0). The maximum flexural strength of 5.36 MPa was achieved by the GMC 0 mix having 10% metakaolin and no replacement of coarse aggregates. This flexural strength of GMC 0 was more than 12% higher than the GC 0 mix, which was the conventional concrete mix. It is also noteworthy that the flexural strengths of all the GMC mixtures were higher compared to their respective GC mixes. As discussed in Section 3.2.2, the filler and pozzolanic effects of metakaolin helped GMC to attain more flexural strength than GC mixtures with the same percentages of coarse aggregate replacements. In the case of GMC mixtures, GMC 30 attained the lowest flexural strength of 3.33 MPa, almost a 38% decrement compared to GMC 0.

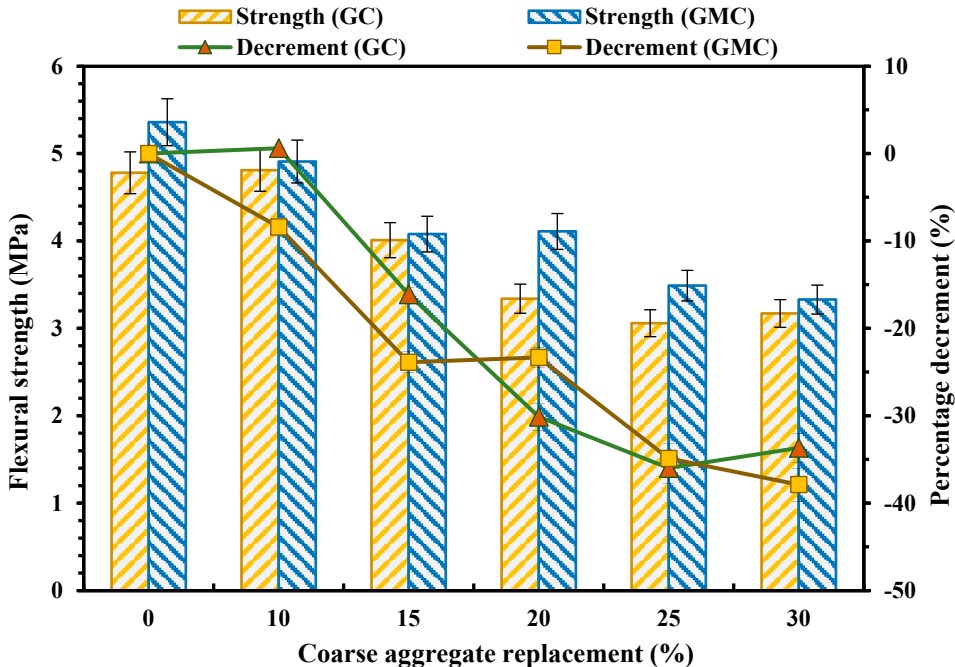

**Figure 10.** Flexural strength of concrete mixes for GC and GMC at 28 days.

### 3.2.6. Relationship between Compressive Strength and Flexural Strength

The interrelationship between the compressive and flexural strength of the concrete incorporating metakaolin and various proportions of glass aggregate at 28 days is illustrated in Figures 11 and 12. The correlation can be used for the flexural strength's anticipation of concrete mixtures incorporating GC and GMC without the need for flexural testing. To forecast flexural strength relying on compressive strength, various standards prescribe different alternative formulas. The ACI 363R, ACI 318, CEB-FIP, and AS 3600 [80–83] standards were employed to determine the flexural strength of concrete with respect to the compressive strength using these prescribed formulas.

$$f_{fr} = 0.94 \sqrt{f_c} \tag{7}$$

$$f_{fr} = 0.62 \sqrt{f_c} \tag{8}$$

$$f_{fr} = 0.46 (f_c)^{2/3} \tag{9}$$

$$f_{fr} = 0.60 \sqrt{f_c} \tag{10}$$

where, $f_{fr}$ = Flexural strength (MPa), $f_c$ = Compressive strength (MPa).

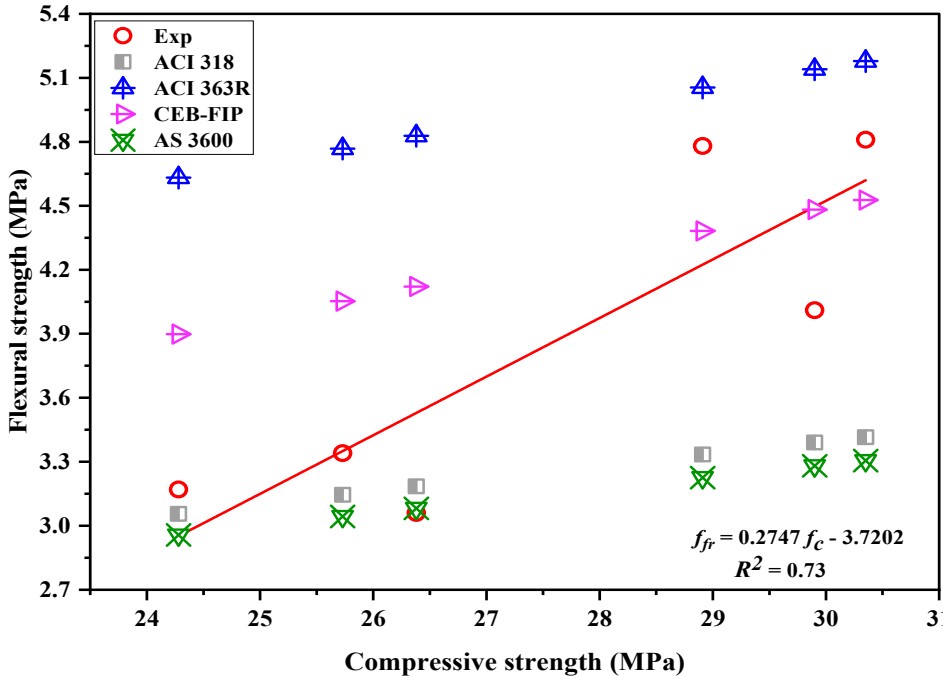

**Figure 11.** Relationship between the compressive and flexural strength of GC mix.

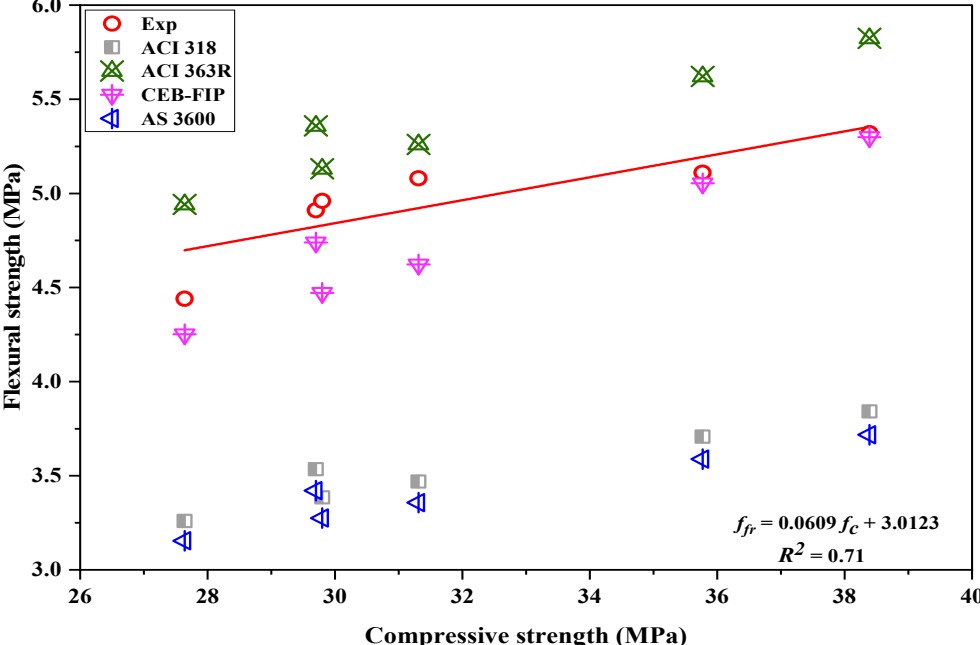

**Figure 12.** Relationship between the compressive and flexural strength of GMC mix.

It can be depicted from Figure 11 that amongst all standards, the estimated values of ACI 363R deviated highest compared to the experimental values for GC mixes. The values according to ACI-318 and AS 3600 lie under the experimental values and deliver a conservative result. A correlation was determined between the compressive and flexural strength with a coefficient of determination of 73%, which indicates a decent relationship

for GC mixes. From the investigation, the following formula (Equation (11)) can be advised for expressing the relationship between the flexural strength and the compressive strength of GC.

$$f_{fr} = 0.2747\, f_c - 3.7202 \tag{11}$$

where, $f_{fr}$ = Flexural strength (MPa), $f_c$ = Compressive strength (MPa).

Figure 12 illustrates the relationship between compressive strength and flexural strength for GMC mixes compared with different standards. It can be observed that the analytical data for all the standards lie under the experimental values except for ACI-363R, which indicates a conservative relationship. According to the study, the relationship between the flexural strength and the compressive strength of GMC may be expressed using the following formula (Equation (12)).

$$f_{fr} = 0.0609\, f_c + 3.0123 \tag{12}$$

where, $f_{fr}$ = Flexural strength (MPa), $f_c$ = Compressive strength (MPa).

*3.3. Sustainability Analysis*

The entire procedure of making concrete, from raw material collection and processing towards the concrete mixture formation, is included inside the system boundary. All activities that fall inside the border include cement production, aggregate collection, and material transportation. The concrete plant's water and power sources are outside the border. The next section provides justification for the system boundary assumptions used.

- 1 $m^3$ is the functional unit, meaning that each mixture yields 1 $m^3$.
- Every mix proportion uses the same concrete processing parameters; hence they are irrelevant to this comparative analysis. The framework includes the necessary processing information for materials.
- Any emissions allocated to the materials from steel or electricity production are not seen as relevant in terms of concrete production and have not been included inside the system boundary. Only processing and transportation emissions are allocated to the materials for this assessment.
- It is optimistically predicted that all transportation will be done via roadway.

Table 6 shows the considered unit cost, embodied energy, and eCO$_2$ emissions of each material for the cost and sustainability analysis. The cost is estimated in MYR (1 MYR = 0.22 USD) currency. The authors have taken the production cost of concrete materials from recent research that complies with the market standards [84]. The cost data of metaklaolin was collected from the manufacturer. The production cost of glass cullet was considered zero as it was collected from waste. However, the transportation of waste glass led to an insignificant cost associated with it. Moreover, the unit cost of production and transportation was assessed with respect to the Malaysian currency. It is to be noted that the cost of concrete materials can vary from market to market. The material embodied energy and eCO$_2$ emission rates for cement, natural coarse aggregate (NCA), and waste glass coarse aggregate (GCA) were collected from Hammond et al. [85]; for fine aggregate (FA) and water were collected from Datta et al. [86] and Bostanci [87], respectively. In the case of metakaolin, eCO$_2$ emission data were gathered from Maddalena et al. [88]. The carbon emissions due to materials transportation were estimated by taking into account the travel distance from pulling out to the production site collaborating with the supplier company.

**Table 6.** Unit cost and eCO$_2$ emission of concrete materials.

| Material | Unit Cost (MYR/kg) | | Material Embodied Energy (MJ/kg) | eCO$_2$ Emission (kg CO$_2$/kg) | | |
| --- | --- | --- | --- | --- | --- | --- |
| | Production | Transportation | | Production | Transportation | Total |
| Cement | 0.378 [84] | 0.01428 | 5.5 [85] | 0.95 [85] | 0.02 | 0.971 |
| Metakaolin | 0.42 | 0.0126 | 3.48 [88] | 0.218 [88] | 0.018 | 0.236 |
| NCA | 0.21 [84] | 0.01092 | 0.083 [85] | 0.005 [85] | 0.021 | 0.026 |
| GCA | - | 0.0084 | 0.052 [85] | 0.0038 [85] | 0.0005 | 0.0043 |
| FA | 0.09114 [84] | 0.00924 | 0.08 [86] | 0.0048 [86] | 0.16 | 0.1648 |
| Water | 0.003696 [84] | - | 0.0009 [87] | 0.00155 [87] | - | 0.00155 |

3.3.1. Effect of Metakaolin and Crushed Glass Aggregates on the Concrete Production Cost

This study conducted a cost analysis of per m$^3$ volume of GC and GMC mixtures as well as the cost of each material used in this study. Table 7 shows the breakdown of costings and the total cost of producing 1 m$^3$ of concrete. From Table 7, it can be depicted that the production cost of GC 0 was the highest at 454.91 MYR. The production cost per m$^3$ of concrete decreased gradually with the incorporation of both the waste glass and metakaolin. However, the incorporation of only glass aggregate to replace natural coarse aggregate reduced the cost to the lowest point of 391.85 MYR. This reduction of concrete cost using a 30% glass cullet as a replacement of NCA was almost 14% compared to conventional concrete. On the other hand, the fabrication cost of GMC 30 containing 30% glass cullet and 10% metakaolin was determined to be 393.51 MYR, which was a 13.5% reduction in cost. Cement and NCA are the two most costly materials among the traditional unreinforced concrete. Replacing these two materials with low-cost new metakaolin and glass cullet significantly reduced the cost.

**Table 7.** Cost assessment of per m$^3$ concrete mixtures.

| Mix ID | Cost of Materials (MYR/m$^3$ Concrete) | | | | | | Total Cost (MYR/m$^3$ Concrete) | Total Cost (USD/m$^3$ Concrete) |
| --- | --- | --- | --- | --- | --- | --- | --- | --- |
| | Cement | Metakaolin | NCA | GCA | FA | Water | | |
| GC 0 | 154.95 | - | 218.49 | - | 80.81 | 0.66 | 454.91 | 97.41 |
| GC 10 | 154.95 | - | 196.64 | 0.83 | 80.81 | 0.66 | 433.89 | 92.91 |
| GC 15 | 154.95 | - | 185.72 | 1.25 | 80.81 | 0.66 | 423.38 | 90.66 |
| GC 20 | 154.95 | - | 174.79 | 1.66 | 80.81 | 0.66 | 412.87 | 88.41 |
| GC 25 | 154.95 | - | 163.87 | 2.07 | 80.81 | 0.66 | 402.36 | 86.16 |
| GC 30 | 154.95 | - | 152.94 | 2.49 | 80.81 | 0.66 | 391.85 | 83.91 |
| GMC 0 | 139.46 | 17.15 | 218.49 | - | 80.81 | 0.66 | 456.57 | 97.77 |
| GMC 10 | 139.46 | 17.15 | 196.64 | 0.83 | 80.81 | 0.66 | 435.55 | 93.27 |
| GMC 15 | 139.46 | 17.15 | 185.72 | 1.25 | 80.81 | 0.66 | 425.04 | 91.01 |
| GMC 20 | 139.46 | 17.15 | 174.79 | 1.66 | 80.81 | 0.66 | 414.53 | 88.76 |
| GMC 25 | 139.46 | 17.15 | 163.87 | 2.07 | 80.81 | 0.66 | 404.02 | 86.51 |
| GMC 30 | 139.46 | 17.15 | 152.94 | 2.49 | 80.81 | 0.66 | 393.51 | 82.26 |

Note: Total costs of mixtures were estimated using the unit costs of materials detailed in Table 6.

Additionally, to ensure comprehensive evaluation, it is imperative to consider the expenses associated with generating 1 MPa of strength for every mixture when conducting a thorough cost-effectiveness assessment [89]. The cost to produce a 1-MPa compressive strength at 60 days, which can also be termed as the cost index of the mixes, is shown in Figure 13, which also depicts the reduction percentages. The cost index of concrete mix can be derived utilizing Equation (13).

$$CI = \frac{Cost}{C_s} \tag{13}$$

where, *CI* = cost index; *Cost* = cost of producing 1 m$^3$ concrete mixture (MYR); and *C$_s$* = compressive strength attained by the mix after 60 days (MPa).

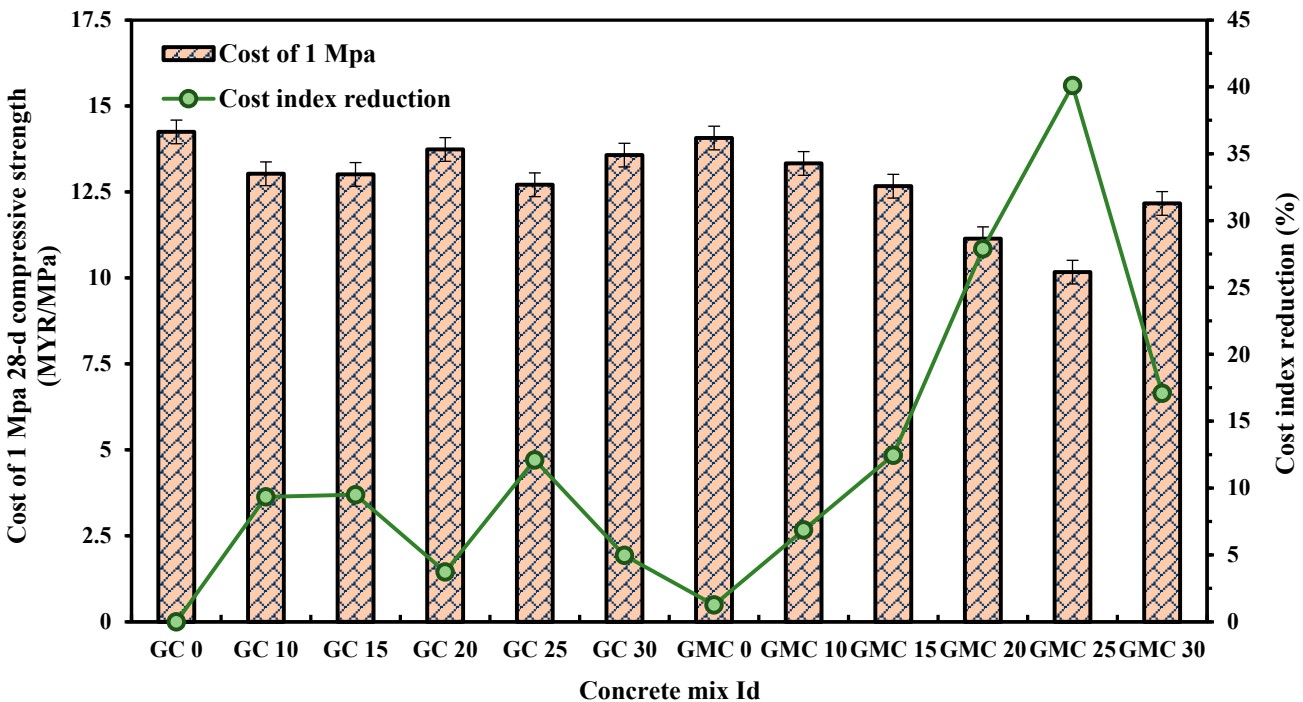

**Figure 13.** Cost index of concrete mixes.

While the control mix showed a cost index of 14.25 MYR/MPa, the inclusion of glass cullet and metakaolin reduced it for all the mixes. The lowest cost for 1 MPa of 10.17 MYR was exhibited by the mix containing 25% glass cullet and 10% metakaolin (GMC 25), an almost 40% reduction, as compared to the control mix. In addition, 20% glass cullet and the same concentration of metakaolin reduced the cost index by a margin of 28%. Furthermore, among the mixes with only glass cullet inclusion, 25% dosage reduced this index by the largest margin, more than a 12% reduction. As the two batches of mixes were designed to substitute natural coarse aggregate and cement, the two most costly materials in concrete, this significantly reduced the cost of mixtures. The design attained satisfactorily and, in some cases, enhanced compressive strength; this significantly reduced the cost of producing 1 MPa, indicating a potential usage of these mixes when the strength limit and serviceability criteria are met.

### 3.3.2. Effect of Metakaolin and Crushed Glass Aggregates on the $eCO_2$ Emission of Concrete

The equivalent carbon dioxide emission during the construction and transportation phase is represented in Table 8. It can be said that the conventional concrete mix (GC 0) with no MK and coarse aggregate replacement emitted maximum $CO_2$, which was 542.2 kg per $m^3$ of concrete volume. Replacement of only natural coarse aggregate with glass cullet aggregate reduced the carbon dioxide emission by only 1.2% with 30% glass cullet aggregate. However, incorporating both glass cullet and metakaolin significantly decreased the amount of carbon dioxide emission. For GMC 30 mixture containing 30% glass cullet and 10% metakaolin, it reduced the carbon dioxide emission to 506.73 kg/$m^3$ concrete, which was almost 8% lower compared to the conventional concrete mix. Using low-carbon emitting material such as metakaolin to replace cement with a high carbon emission rate can be attributed to this reduction in $eCO_2$ emission value.

**Table 8.** Equivalent $CO_2$ emission of per $m^3$ concrete mixtures.

| Mix ID | eCO₂ Emission (kg CO₂/m³ Concrete) | | | Percentage of CO₂ Emission | |
|---|---|---|---|---|---|
| | Production | Transportation | Total | Production (%) | Transportation (%) |
| GC 0 | 384.33 | 157.86 | 542.20 | 70.88 | 29.12 |
| GC 10 | 384.22 | 155.84 | 540.05 | 71.144 | 28.86 |
| GC 15 | 384.16 | 154.82 | 538.98 | 71.27 | 28.73 |
| GC 20 | 384.10 | 153.81 | 537.91 | 71.41 | 28.59 |
| GC 25 | 384.03 | 152.79 | 536.83 | 71.54 | 28.46 |
| GC 30 | 383.98 | 151.78 | 535.76 | 71.67 | 28.33 |
| GMC 0 | 355.42 | 157.75 | 513.17 | 69.26 | 30.74 |
| GMC 10 | 355.30 | 155.72 | 511.02 | 69.53 | 30.47 |
| GMC 15 | 355.24 | 154.7 | 509.95 | 69.66 | 30.34 |
| GMC 20 | 355.18 | 153.69 | 508.87 | 69.79 | 30.2 |
| GMC 25 | 355.12 | 152.68 | 507.80 | 69.93 | 30.07 |
| GMC 30 | 355.06 | 151.66 | 506.73 | 70.07 | 29.93 |

The assessment of environmental impact may involve considering the eco-strength efficiency of concrete as an additional metric for evaluation. The concept of eco-strength efficiency, as mentioned by Alnahhal et al. [90], and referred to as $CO_2$ intensity by Damineli et al. [91], pertains to the amount of $CO_2$ emissions generated per unit of performance. The calculation of this parameter was derived utilizing Equation (14).

$$C_i = \frac{CO_2}{C_s} \tag{14}$$

where $C_i$ = eco-strength efficiency, which denotes the intensity of $CO_2$ emissions; $CO_2$ = carbon dioxide emissions by the concrete mixes (kg $CO_2$/$m^3$ concrete), as determined through the data provided in Table 5; and $C_s$ = compressive strength attained by the mix after 60 days (MPa).

To ensure a systematic and consistent comparison of the mixes, the eco-strength efficiency of different combinations is juxtaposed with their respective compressive strength. The outcomes of the analysis are illustrated in Figure 14, where the compressive strength values at the 60-day mark are visually portrayed by the bar chart, whereas the $CO_2$ intensity is graphically presented as a line along the secondary axis. The $CO_2$ intensity enables the assessment of both the effectiveness and the role of concrete mixes in contributing to Global Warming Potential based on their unit strength, positioning it as a reliable indicator for estimating the environmental impact of concrete utilization [91]. In the majority of instances, the $CO_2$ concentration tends to increase as the quantity of Portland cement in the concrete mixture rises to enhance the compressive strength as per requirements. However, by partially replacing natural aggregate and cement by a waste glass cullet and metakaolin, it becomes possible to decrease the $CO_2$ intensity while maintaining the desired strength level, in fact, enhancing it by a noticeable margin. All the designed mixes for the current study, except the GC 20 and GC 30 mixes, have exhibited reduced carbon intensity. Among these, the incorporation of 25% glass cullet and 10% metakaolin resulted in the most significant reduction by 33%, shifting the intensity from 16.98 MPa/kg $CO_2 \cdot m^{-3}$ for the control mix to 12.78 MPa/kg $CO_2 \cdot m^{-3}$ for GMC 25 mix. As the production process of substituting materials (i.e., glass cullet and metakaolin) emit significantly lower levels of $CO_2$ than the substituted stone aggregate and cement, as well as impart enhanced compressive strength, the carbon intensity was reduced. This outcome indicates that there is potential for these materials to be used in the production of sustainable structural concrete without compromising its strength.

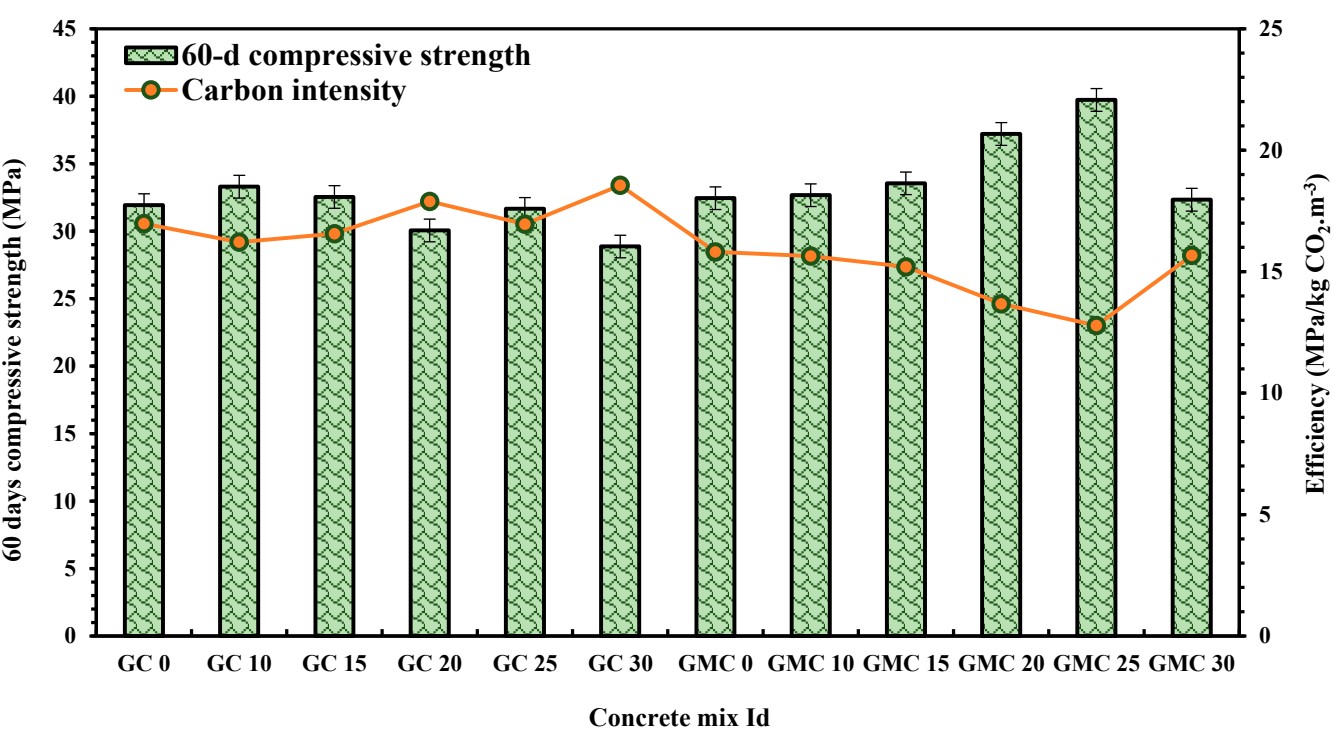

**Figure 14.** Eco-strength efficiency with respect to compressive strength.

## 4. Conclusions

Based on existing experimental outcomes, the following significant observations and conclusions can be reached:

- The overall workability of the concrete mixtures incorporating glass aggregate increased gradually, both for the GC and GMC. The smooth surface and weaker cohesive bonding of the crushed glass aggregates enhanced the overall workability. On the other hand, the decelerating trend of MK increased the workability and moveability even further.

- The optimum compressive strength was achieved with GC 10 for only glass aggregate incorporated concrete, which was 5% higher than the control mix at 28 days. Increased GC content reduced the compressive strength due to their flat and flaky shape, which can alter the aggregate matrix and induce inhomogeneity.

- The incorporation of 10% metakaolin into the glass concrete mix accelerated the compressive strength increment. The pozzolanic action of MK creates secondary C-S-H gels and micro-filler effect reduces pore sizes. As a result, a 30% increment of compressive strength was observed for GMC 25.

- The flexural strength results were identical to those of the compressive strength tests, where the incorporation of MK augmented in higher strengths. According to these determinations, concrete incorporating 10% glass aggregate possessed optimum flexural strength of 4.81 MPa for GC mixes. On the other hand, GMC 0 with 10% metakaolin attained the maximum flexural strength of 5.36 MPa.

- The attained compressive strength for both the GC and GMC mixes was compared with different standards like ACI 318, ACI 363R, AS 3600, and CEB-FIP to determine analytical tensile strength values without performing experiments. The outcomes of the comparison were plotted in a scattered manner where all the standards except ACI 363R exhibited conservative results.

- The outcomes of the relation between flexural strength and compressive strength tests revealed a decent linear connection. The outcomes were also calculated with different standards, and a conservative correlation was found for both the GC and GMC mixes.

- Among all mix proportions, the GMC 25 produced a satisfactory optimal performance in terms of rheological and mechanical characteristics. It is feasible to generate high-strength concrete by employing the crushed glass aggregate alongside metakaolin incorporation as an additional cementing element which leads to waste manufacture modification and a sustainable alternative for the concrete industry.
- The production cost and equivalent carbon dioxide emission by concrete can respectively be reduced by up to 13.5% and 8% with the incorporation of glass cullet and metakaolin at optimal concentrations of 25% and 10%.
- The cost index and carbon index can respectively be reduced by 40% and 33% with the optimum dosages of GC and MK, establishing that glass cullet and metakaolin can be termed as sustainable building materials and reduce the carbon footprint of the concrete industry.

The present study considered untreated waste glass cullet and metakaolin as partial replacements for stone aggregate and cement content for producing normal concrete and revealed that both these materials can be used in isolated or combined to produce low-cost concrete with similar mechanical strengths. Future studies should consider other supplementary cementitious materials, such as fly ash, glass powder, quartz powder, limestone powder, etc., along with waste glass cullet aggregates to produce eco-friendly sustainable concrete. In addition, various extraction and treatment methods [92,93] for these waste materials should be considered for future studies. Furthermore, extensive exploration in terms of microstructural characterization of glass cullet and metakaolin concrete needs to be conducted in future investigations.

**Author Contributions:** N.M.S.H.: Conceptualization, Methodology, Data curation, Supervision, Project administration, Writing—original draft. N.M.N.S.: Investigation, Data curation, Writing—original draft. M.H.R.S.: Investigation, Writing—review and editing. M.M.M.: Investigation, Writing—review and editing. M.S.I.: Investigation, Data curation, Writing—review and editing. M.J.M.: Methodology, Validation, Writing—review and editing. All authors have read and agreed to the published version of the manuscript.

**Funding:** This research received no external funding.

**Institutional Review Board Statement:** Not applicable.

**Informed Consent Statement:** Not applicable.

**Data Availability Statement:** The data supporting these findings are available within the article.

**Acknowledgments:** The structural properties of concrete were analyzed at the Heavy Structural and Materials laboratory of Bandar Universiti Terknologi Legenda Malaysia (University of East London Malaysia Branch). All the authors associated with this article showing their utmost gratitude to the technicians, lab-attendant and other associates along with their support during the fabrication of specimens and machine operations. The authors also gratefully acknowledged the personnel associated with the data collections and testing of laboratory specimens as part of their research dissertation/thesis.

**Conflicts of Interest:** The authors declare no conflict of interest.

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
