# Peer review of "Utilization of Waste Glass Cullet as Partial Substitutions of Coarse Aggregate to Produce Eco-Friendly Concrete: Role of Metakaolin as Cement Replacement"

_sustainability, doi:10.3390/su151411254_

Round 1
Reviewer 1 Report
Please see the attached file for comments.

Ok, but may be improved in a several cases.
Author Response
RESPONSE TO REVIEWERS’ COMMENTS
Manuscript ID: sustainability-2460983
Title: Role of waste glass and metakaolin as a partial substitution of coarse aggregate and cement to produce eco-friendly concrete.
Correspondence Author: Md Jihad Miah
Dear Editor,
Dear reviewers,
Thanks for your letter and the reviewers’ comments on our manuscript updated titled “Utilization of waste glass cullet as partial substitutions of coarse aggregate to produce eco-friendly concrete: Role of metakaolin as cement replacement.” The comments are valuable, helpful for improving the manuscript, and even instructive for future research. Accordingly, we have carefully studied these insightful comments and revised our manuscript according to your suggestions to meet the approval of Sustainability. Please find below the corrections and the response to the comments. The reviewers’ comments are copied in black, the corrections and the responses are marked in blue in this report and the revised manuscript, and citations from the revised manuscript are marked in blue in italic.
Reviewer #1: Role of waste glass and metakaolin as a partial substitution of 2 coarse aggregate and cement to produce eco-friendly concrete
Comment #1. Please concise the Abstract
Author’s response: Thank you for your comment. Amended as suggested.
Comment #2. List 2 to 3 environmental effects Line -47 (……………it has a significant 46 environmental effect [2]……)
Author’s response: Thanks for the suggestion. The authors have listed 3 environmental effects. Please refer to Lines 42-45, which are as follows:
“Because of environmental degradation, carbon emissions, and heavy energy usage, it has a range of detrimental environmental effects, including climate change, scarcity of natural resources, and air pollution [2].”
Comment #3. Line -50, Consequently, traditional aggregate resources are deteriorating at an accelerated rate…………… This sentence is not completed!
Author’s response: Thank you for pointing out this. Amended as suggested. Please refer to Lines 47-49, which are as follows:
“Consequently, traditional aggregate resources are deteriorating at an accelerated rate to meet this ever-increasing demand for concrete aggregates.”
Comment #4. Title of section 2.1 (line 141) should be “Research Plan” instead of “Research Method”.
Author’s response: Acknowledged and amended in the revised manuscript.
Comment #5. Fig. 1, Please show one box in the flow diagram at the very end like “conclusion or/ decision” to make the final decision if this material is sustainable/ cost-effective when used as the replacement of the traditional aggregates.
Author’s response: The authors would like to thank the reviewer for this comment to improve the quality of the manuscript. The authors have amended Figure 1 as per the reviewer’s comment.
Comment #6. Please mention the ASTM type of the cement-like Type I/II or something ?
Author’s response: The authors have mentioned the ASTM type of the cement as Type-I mark in the revised manuscript. Please refer to Line 156-159, which are as follows:
“To conduct the present research, the Type I ordinary Portland Cement (OPC) of the “Seladang” brand, manufactured by Tenggara Cement Manufacturing Sdn, was used as a primary binder for the concrete mixes that correspond to the ASTM C150 / C150M-20 [59] standard.”
Comment #7. Table-1, in order to be consistent, please write “Potassium Oxide” instead of “Potash”. Also please input the LOI for cement. If you can’t run a sample, likely you will get it from the manufacturer.
Author’s response: The authors have amended and addressed the LOI in the revised manuscript; please refer to Table 1 in the revised manuscript.
Comment #8. Did you do any abrasion test for glass cullet?
Author’s response: Indeed, the outcomes of the abrasion resistance of glass cullet could give a better explanation of the mechanism of concrete's mechanical properties. The authors agree with the reviewer and appreciate the reviewer's suggestion, which was not considered in this project. However, this ongoing project could be considered in the bond strength and durability properties after exposure to aggressive environments. Thanks.
Comment #9. Please show the particle size distribution of all aggreagtes.
Author’s response: Thank you for your suggestion, and amended as suggested. Please refer to Lines 178-180 and in Figure 3 of the revised manuscript, which is as follows:
“The gradation curve of stone chips as coarse aggregate, crushed glass aggregate, and fine aggregate (sand), is depicted in Figure 3 with upper and lower limits in accordance with ASTM-C33 [62].”
Comment #10. Line 198, check the following sentence……………..mm ×500 mm prisms built in the laboratory…………. “Prisms that were made/prepared in the laboratory”
Author’s response: Amended as suggested; please check lines 207-209 in the revised manuscript, which are as follows:
“Compressive strength was measured using 100 mm × 200 mm cylinder specimens, whereas flexural strength was examined using 100 mm × 100 mm × 500 mm prisms that were prepared in the laboratory.”
Comment #11. Please show the standard deviations of each variable in Fig.-3
Author’s response: Amened as suggested. Please refer to Fig. 4 (which was Fig. 3) in the revised manuscript as shown below:
Comment #12. Table-3 or section 3.2.1, you have done a couple of statistical tests, Did you do any significance tests on compressive strength? Is there any significant mean difference exists among all samples? Otherwise, this analysis will not be complete!
Author’s response: The authors would like to thank the reviewer for this comment to improve the quality of the manuscript. The authors have revised the statistical tests for Table-3 as per the reviewer’s comment; please refer to Line 285. The authors have successively tested three samples for each curing day, and the revised manuscript shows their statistical parameters.
Furthermore, the authors have added a new Section 3.2.3 Analysis of variance (ANOVA) for compressive strength test results, to conduct a complete significace analysis of the compressive strengths as per the reviwer’s suggestion. This analysis showed that both the glass cullet and metakaolin have shown statistical significance in concrete mixtures’ later age strengths, which also stands in line with the experimental outcomes. Please refer to Lines 386-402, which are as follows:
“3.2.3 Analysis of variance (ANOVA) for compressive strength test results
A single factor ANOVA test was performed to assess the statistical significance of GC and GMC mixes on the compressive strength of concrete. The test was initiated for all the curing intervals to observe their statistical significance at a 95% significance level (α= 0.05). The results for both the GC and GMC mix combinations are detailed in Table 5. The null hypothesis states that the percentage of glass cullets in GC and GMC mixes have no statistical significance to their compressive strengths such as, if the P-value is less than 0.05, it can be rejected, which indicates that a statistically significant relationship exists among the two variables. From the test outcomes, it can be seen that at early curing intervals of 3 and 7 days, there is no statistically significant relationship for both the GC and GMC mixes. In contrast, at early curing days of 28 and 60, a significant statistical relationship can be found. This test outcomes are in line with the findings discussed in sections 3.2.1 and 3.2.2, where it was evident that the glass cullet and metakaolin both affected the later age compressive strengths more than their earlier ages.”
Comment #13. Similar comments for Table 4.
Author’s response: Amened as suggested. The authors have added a new Section 3.2.3 Analysis of variance (ANOVA) for compressive strength test results, and Table 5. ANOVA test results for different curing intervals of GC and GMC mixes to their mean compressive strengths.
Comment #14. Please be consistent in writing equations. For instance, you have written (1) and (2) for equations 1 and 2 respectively, while (Eq 3), (Eq 4), and (Eq 5) for equations 3, 4, 5, and 6 respectively, and so on. Also, check equation 13.
Author’s response: Thank you for pointing this out. The authors have revised the manuscript accordingly.
Comment #15. Your equation number 13 is showing something different that you have shown in Figure 11. This is a significant mistake!
ffr = 0.0609 fc– 3.0123 (Eq 12) should it be a negative(-) 3.0123 or positive (+) 3.0123 ???
Author’s response: The authors highly appreciate the reviewer for pointing this out. This is a significant typo mistake. The authors have corrected equation number 12 as it should be positive (+).
Comment #16. You can show the cost by USD ($) beside the MYR, hence it will be easier for the readers.
Author’s response: Acknowledged. The authors have amended the manuscript as suggested. Please refer to Table 7, where the cost in USD was showed with a conversion rate of 1 USD=4.67 MYR.
Comment #17. Write the proper explanation why GC 25 have higher compressive strength than that of GC 20 as shown in Fig. 5. What are the optimum dosages of “GC” as well based on the compressive strength?
Author’s response: Thank you for your comment. The authors have explained why GC25 has higher strength than GC20. Please refer to Lines 294-303, which are as follows:
“Notably, the compressive strength for GC 25 was higher than GC 20 but lower than the control mix. This increase can be attributed to the suitable compaction of the glass aggregate. Moreover, it is possible that the angular shaped aggregates were well distributed and oriented along with the natural stone aggregate. The combined effects can impart a stiffer matrix to the concrete, resulting in higher strength for 25% GC incorporation. Similar irregularities in compressive strength of higher contents of waste glass were observed by Arabi, et al. [69], when they used windshield waste glass aggregate to replace natural coarse aggregate partially. In their study, a notable jump in strength was seen when the glass aggregate was used at a higher dosage along with recycled natural coarse aggregate.”
Moreover, the authors would like to mention that the optimum dosage of GC still remains 10% because this mix attained the maximum strength, which was significantly higher than the control mix. While GC15 also showed slightly higher strength than GC0, all the other mixtures showed reduced strength. Please refer to lines 316-317, which are as follows:
“As a result, 10% replacement of glass aggregate displayed the optimum result.”
Comment #18. Give proper citations for line numbers 288-296.
Author’s response: The authors have provided proper citations for the lines mentioned. Please refer to Lines 306-316 in the revised manuscript, which is as follows:
“Indeed, the flat and flaky shape of glass aggregates is more vulnerable to break/damage (i.e., easy to break with lower load due to the higher slenderness effect) when they experience the mechanical load, which also depends on their orientation [70]. Also, the glass aggregate has a smooth surface, which can create a poor interfacial transition zone (ITZ) around the glass aggregates and cement paste [71]. Last but not least, glass aggregate has significantly high brittleness, a great disadvantage of glass particles. Hence, when the concrete experience the mechanical load, the cracks can easily initiate at the ITZ, and then the cracks propagate more effortlessly as the glass aggregates are more vulnerable to detach from the cement mortar, i.e., failure may occur primarily due to the initiation of cracks at ITZ rather than the glass and stone aggregate fracture [72].”
Comment #19. Giver a proper explanation why GMC 15 has higher compressive strength at an early age of 7 days while GMC 25 have higher compressive strength at 60 days.
Author’s response: The authors have amended the manuscript as suggested. Please refer to lines 378-385, which are as follows:
“Moreover, the significant jump in the strength of GMC 25 concrete mix can be attributed to the hydration heat of MK based concrete. At early ages, hydration heat can affect the compressive strength of concrete by increasing shrinkage and thermal cracks [78]. These effects can be intensified using MK, a high reactivity pozzolan [79]. Along with this, as discussed in Section 3.1.1, incorporating glass aggregate increases the workability and makes more water available for the hydration process. As a result, the higher dosage of GC exhibited comparatively lower strengths at the early ages of 3 and 7 days, whereas higher strengths at later ages of 28 and 60 days.”

Reviewer 2 Report
Paper ID: sustainability-2460983
Type:Article
Title: Role of waste glass and metakaolin as a partial substitution of coarse aggregate and cement to produce eco-friendly concrete
This paper investigates waste glass and metakaolin as a partial substitution of coarse aggregate and cement to produce concrete. Although I appreciate the efforts of the authors, this work is not eligible for a reputable journal such as Sustainability (IF:3.89-Q2). There is nothing new about what is known in the literature. The title of the paper is “Role of waste glass and metakaolin as a partial substitution of coarse aggregate and cement to produce eco-friendly concrete”. Metakaolin is a coarse aggregate ?. Some relevant articles presented in the literature are presented below:
Guignone, G. C., Vieira, G. L., Zulcão, R., Degen, M. K., de Moraes Mittri, S. H., & Baptista, G. (2020). Incorporation of glass powder and metakaolin as cement partial replacement to improve concrete mechanical properties and increase service life. Journal of Composite Materials, 54(21), 2965-2983.
Ouldkhaoua, Y., Benabed, B., Abousnina, R., Kadri, E. H., & Khatib, J. (2020). Effect of using metakaolin as supplementary cementitious material and recycled CRT funnel glass as fine aggregate on the durability of green self-compacting concrete. Construction and Building Materials, 235, 117802.
Chand, G., Happy, S. K., & Ram, S. (2021). Assessment of the properties of sustainable concrete produced from quaternary blend of portland cement, glass powder, metakaolin and silica fume. Cleaner Engineering and Technology, 4, 100179.
Si, R., Dai, Q., Guo, S., & Wang, J. (2020). Mechanical property, nanopore structure and drying shrinkage of metakaolin-based geopolymer with waste glass powder. Journal of Cleaner Production, 242, 118502.
Author Response
RESPONSE TO REVIEWERS’ COMMENTS
Manuscript ID: sustainability-2460983
Title: Role of waste glass and metakaolin as a partial substitution of coarse aggregate and cement to produce eco-friendly concrete.
Correspondence Author: Md Jihad Miah
Dear Editor,
Dear reviewers,
Thanks for your letter and the reviewers’ comments on our manuscript updated titled “Utilization of waste glass cullet as partial substitutions of coarse aggregate to produce eco-friendly concrete: Role of metakaolin as cement replacement.” The comments are valuable, helpful for improving the manuscript, and even instructive for future research. Accordingly, we have carefully studied these insightful comments and revised our manuscript according to your suggestions to meet the approval of Sustainability. Please find below the corrections and the response to the comments. The reviewers’ comments are copied in black, the corrections and the responses are marked in blue in this report and the revised manuscript, and citations from the revised manuscript are marked in blue in italic.
Reviewer# 2:
This paper investigates waste glass and metakaolin as a partial substitution of coarse aggregate and cement to produce concrete. Although I appreciate the efforts of the authors, this work is not eligible for a reputable journal such as Sustainability (IF:3.89-Q2). There is nothing new about what is known in the literature. The title of the paper is “Role of waste glass and metakaolin as a partial substitution of coarse aggregate and cement to produce eco-friendly concrete”. Metakaolin is a coarse aggregate ?. Some relevant articles presented in the literature are presented below:
Guignone, G. C., Vieira, G. L., Zulcão, R., Degen, M. K., de Moraes Mittri, S. H., & Baptista, G. (2020). Incorporation of glass powder and metakaolin as cement partial replacement to improve concrete mechanical properties and increase service life. Journal of Composite Materials, 54(21), 2965-2983.
Ouldkhaoua, Y., Benabed, B., Abousnina, R., Kadri, E. H., & Khatib, J. (2020). Effect of using metakaolin as supplementary cementitious material and recycled CRT funnel glass as fine aggregate on the durability of green self-compacting concrete. Construction and Building Materials, 235, 117802.
Chand, G., Happy, S. K., & Ram, S. (2021). Assessment of the properties of sustainable concrete produced from quaternary blend of portland cement, glass powder, metakaolin and silica fume. Cleaner Engineering and Technology, 4, 100179.
Si, R., Dai, Q., Guo, S., & Wang, J. (2020). Mechanical property, nanopore structure and drying shrinkage of metakaolin-based geopolymer with waste glass powder. Journal of Cleaner Production, 242, 118502.
Author’s response: The authors highly appreciate the reviewer’s comment. However, the authors believe that the title of the manuscript has raised confusion about the objective of this study. This study first examined the effect of partially replacing natural coarse aggregate with glass cullets at 10, 15, 20, 25, and 30% in concrete. Then it evaluated the effect of using the same proportions of glass cullet to replace natural coarse aggregate along with 10% metakaolin as a partial replacement of cement content in concrete. Thus, glass cullets were used to replace coarse aggregate, and metakaolin was used to replace cement.
The novelty of this paper lies in the incorporation of glass cullets as coarse aggregate, as glass dust is already established as a suitable replacement for fine aggregate, which the reviewer also mentioned. The authors have refined section “1.1. Research Significance” to highlight the novelty of this paper. Please refer to lines 135-140, which are as follows:
“Previously, numerous studies have been conducted on improving the performance of concrete made with fine glass aggregate and dust. However, only a few investigations have been done on the utilization of waste glass cullet as a coarse aggregate. In addition, employing glass aggregate from domestically supplied waste glass bottles as coarse aggregate along with Metakaolin has not been initiated yet.”
Furthermore, the authors have changed the title of the manuscript from “Role of waste glass and metakaolin as a partial substitution of coarse aggregate and cement to produce eco-friendly concrete” to “Utilization of waste glass cullet as partial substitutions of coarse aggregate to produce eco-friendly concrete: Role of metakaolin as cement replacement” for better representation of the study.
Moreover, the authors believe that the outcomes of this study will have a significant scientific impact and could be used to develop the design guidelines for waste glass cullet aggregate concrete. These findings are also expected to assist engineers in utilizing waste glass cullet to partially replace natural aggregate to produce low and medium-strength concrete that commonly use in the rural area of many countries in Asia. This will also reduce the demand for new natural aggregate, contribute to the circular economy, solve dumping issues, and save the environment. Using waste glass cullet in concrete can reduce several Mt CO2/year from the atmosphere, which satisfies sustainability. It is acknowledged that more data would be good to have better correlations. Continuous research conducted by the authors will try to emphasize this issue in the future.

Reviewer 3 Report
The article investigates the attributes of concrete incorporating glass waste and metakaolin as substitutes for coarse aggregate and cement respectively. A lot of work has been done The authors are suggested to further revise the manuscript. Some suggestions:
1. Line 95, make a brief explanation of what “proper pozzolanic reaction” is.
2. Line 180, “The water-cement ratio in such concrete mixtures was set at 0.56“, but the water-cement ratio in Table 2 is not 0.56.
3. What is the source of the horizontal coordinate data in Figures 7, 8, 10 and 11?
4. In section 3.3, for the calculation of the sustainability analysis, please indicate the boundary of the calculation at the beginning.
5. Please indicate the source of the cost data in Table 6, just as in Table 5
6. Line 452-455, list the equation for the cost index as carbon index in Equation 13.
7. Standardize the format of all equations in the article (equations 1, 2 and 13 are not consistent with the others); besides, there are no units in the notes for equation 13.
8. Refine the conclusion section so that it corresponds to the previous article section.
Author Response
RESPONSE TO REVIEWERS’ COMMENTS
Manuscript ID: sustainability-2460983
Title: Role of waste glass and metakaolin as a partial substitution of coarse aggregate and cement to produce eco-friendly concrete.
Correspondence Author: Md Jihad Miah
Dear Editor,
Dear reviewers,
Thanks for your letter and the reviewers’ comments on our manuscript updated titled “Utilization of waste glass cullet as partial substitutions of coarse aggregate to produce eco-friendly concrete: Role of metakaolin as cement replacement.” The comments are valuable, helpful for improving the manuscript, and even instructive for future research. Accordingly, we have carefully studied these insightful comments and revised our manuscript according to your suggestions to meet the approval of Sustainability. Please find below the corrections and the response to the comments. The reviewers’ comments are copied in black, the corrections and the responses are marked in blue in this report and the revised manuscript, and citations from the revised manuscript are marked in blue in italic.
Reviewer #3:
The article investigates the attributes of concrete incorporating glass waste and metakaolin as substitutes for coarse aggregate and cement respectively. A lot of work has been done The authors are suggested to further revise the manuscript. Some suggestions:
Comment #1. Line 95, make a brief explanation of what “proper pozzolanic reaction” is.
Author’s response: Thank you for your suggestion. Amended as suggested; please refer to Lines 97-102, which are as follows:
“The released silica, from glass dissolution, reacts with calcium hydroxide (CH) to form C-(N)-S-H (alkali-silica gel) with different compositions depending on the system. This pozzolanic reaction of glass not only consumes portlandite to form in-situ C-S-H, which appears as reaction rim around glass grains, and precipitated C-S-H but also reduces monosulfate level, which in turn, augmented in enhanced mechanical properties. ”
Comment #2. Line 180, “The water-cement ratio in such concrete mixtures was set at 0.56“, but the water-cement ratio in Table 2 is not 0.56.
Author’s response: Thank you for your observation and comment, and the authors agree with the reviewer. This is a typo mistake. The adopted water-cement ratio for this study was 0.45 across all mix combinations, as reported in Table 2. The statement has been corrected in the revised manuscript; please refer to Lines 190-192, whihs are as follows:
“The water-cement ratio in such concrete mixtures was set at 0.45, and the compressive strength target was set at 30 MPa after 28 days.”
Comment #3. What is the source of the horizontal coordinate data in Figures 7, 8, 10 and 11?
Author’s response: The source of the horizontal coordinate data in Figures 8 and 11 in the revised manuscript is Table 3. Similarly, the source for Figures 9 and 12 in the revised manuscript is Table 4. The authors have considered the 28 days compressive strengths for all the mix combinations as their specified compressive strength. The authors used these data to predict the analytical split tensile strengths for GC (Figure 8) and GMC (Figure 9) mixes. The authors also considered the data gathered from laboratory experiments on the flexural strength to establish a relationship between the concrete mix’s 28 days compressive strength and flexural strength for both the GC mixes (Figure 11) and GMC mixes (Figure 12).
Comment #4. In section 3.3, for the calculation of the sustainability analysis, please indicate the boundary of the calculation at the beginning.
Author’s response: The authors have amended the reviewer’s comment in the revised manuscript. Please refer to lines 477-491, which are as follows:
“The entire procedure of making concrete, from raw materials collection and processing towards the concrete mixture formation, is included inside the system boundary. All activities that fall inside the border include cement production, aggregate collection, and material transportation. The concrete plant's water and power sources are outside the border. The next section provides justification for the system boundary assumptions used.
- 1 m3 is the functional unit, meaning that each mixture yields 1 m3.
- Every mix proportion uses the same concrete processing parameters; hence they are irrelevant to this comparative analysis. The framework includes the necessary processing information for materials.
- Any emissions allocated to the materials from steel or electricity production are not seen as relevant in terms of concrete production and have not been included inside the system boundary. Only processing and transportation emissions are allocated to the materials for this assessment.
- It's optimistically predicted that all transportation will be done via roadway.”
Comment #5. Please indicate the source of the cost data in Table 6, just as in Table 5.
Author’s response: The authors appreciate the reviewer's comment. The authors have considered the standard cost data for all the concrete materials used in a recent journal article conducted by Dr. Habibur Rahman Sabuz (an author in the current study). Please refer to Lines 493-497, which are as follows:
“The authors have taken the production cost of concrete materials from recent research that complies with the market standards [84]. The cost data of metaklaolin was collected from the manufacturer. The production cost of glass cullet was considered zero as it was collected from waste. However, the transportation of waste glass led to an insignificant cost associated with it.”
As per the reviewer’s suggestion, the authors also added the source of cost data as a table footer in Table 7; please refer to line 522:
“Note: Total costs of mixtures were estimated using the unit costs of materials detailed in Table 6.”
Comment #6. Line 452-455, list the equation for the cost index as carbon index in Equation 13.
Author’s response: Thanks for the suggestion. The authors have revised the manuscript accordingly. Please refer to lines 527-530 and Equation (13) in the revised manuscript, which is as follows:
“The cost index of concrete mix can be derived utilizing Equation (13).
|
(13) |
where, CI = cost index; Cost = cost of producing 1m3 concrete mixture (MYR); and Cs = compressive strength attained by the mix after 60 days (MPa).”
Comment #7. Standardize the format of all equations in the article (equations 1, 2 and 13 are not consistent with the others); besides, there are no units in the notes for equation 13.
Author’s response: The authors appreciated this observation by the reviewer and standardized the format of all equations. The authors have amended the comment by stating the units in Equation 13. Please refer to lines 563-566 in the revised manuscript, which is as follows:
“where, Ci = eco-strength efficiency, which denotes the intensity of CO2 emissions; CO2 = carbon dioxide emissions by the concrete mixes (kg CO2/m3 concrete), as determined through the data provided in Table 5; and Cs = compressive strength attained by the mix after 60 days (MPa).”
Comment #8. Refine the conclusion section so that it corresponds to the previous article section.
Author’s response: Thanks for the reviewer's recommendation. Amended as suggested; please refer to Section 4 Conclusions in the revised manuscript.

Reviewer 4 Report
The article titled "Role of waste glass and metakaolin as a partial substitution of coarse aggregate and cement to produce eco-friendly concrete" discusses the utilization of waste products as a vital aspect of the construction industry to safeguard environmental assets and mitigate pollution, as well as to achieve long-term sustainable development. The study covers some parameters to evaluate the product. This study is thorough and comprehensive. However, there are some points that the authors should address before the acceptance and publication.
1. Authors should highlight the novelty of using waste glass and metakaolin that has not illustrated in previous studies with comparison.
2. The introduction requires more improvement by including more background about the partial substitution of various wastes supported by more recent literatures such as biological waste polymeric waste. These two types of wastes could be useful in current study; 10.1016/j.pnucene.2020.103285 and 10.1016/j.cscm.2021.e00664
3. In current study, many parameters have been evaluated, what about the porosity of the produced composite?
4. Spectroscopic investigations such as FTIR, XRD, SEM could add more important information about the proposed mortar.
5. The discussion needs more deep investigations.
6. What's the research significance of this paper? What new insights do you provide? The authors are invited to provide in details.
7. In conclusion, I believe that the theme of this manuscript can be consistent generally with the theme of Sustainability and specially with the SI: Recent Developments on the Use of Sustainable Retrofitting and Construction Materials.
8. At the same time, the manuscript needs to be improved and the authors should edit the manuscript in accordance with the guidelines mentioned above.
Author Response
RESPONSE TO REVIEWERS’ COMMENTS
Manuscript ID: sustainability-2460983
Title: Role of waste glass and metakaolin as a partial substitution of coarse aggregate and cement to produce eco-friendly concrete.
Correspondence Author: Md Jihad Miah
Dear Editor,
Dear reviewers,
Thanks for your letter and the reviewers’ comments on our manuscript updated titled “Utilization of waste glass cullet as partial substitutions of coarse aggregate to produce eco-friendly concrete: Role of metakaolin as cement replacement.” The comments are valuable, helpful for improving the manuscript, and even instructive for future research. Accordingly, we have carefully studied these insightful comments and revised our manuscript according to your suggestions to meet the approval of Sustainability. Please find below the corrections and the response to the comments. The reviewers’ comments are copied in black, the corrections and the responses are marked in blue in this report and the revised manuscript, and citations from the revised manuscript are marked in blue in italic.
Reviewer #4:
The article titled "Role of waste glass and metakaolin as a partial substitution of coarse aggregate and cement to produce eco-friendly concrete" discusses the utilization of waste products as a vital aspect of the construction industry to safeguard environmental assets and mitigate pollution, as well as to achieve long-term sustainable development. The study covers some parameters to evaluate the product. This study is thorough and comprehensive. However, there are some points that the authors should address before the acceptance and publication.
Comment #1. Authors should highlight the novelty of using waste glass and metakaolin that has not illustrated in previous studies with comparison.
Author’s response: The authors have rewritten the section “1.1 Research Significance” to highlight the novelty of incorporating waste glass cullet and metakaolin in comparison to previous studies. Please refer to lines 135-140, which are as follows:
“Previously, numerous studies have been executed on the improving performance of concrete made with fine glass aggregate and glass dust. However, only a few investigations have been done on the utilization of waste glass cullet as a coarse aggregate. In addition, employing glass aggregate from domestically supplied waste glass bottles as coarse aggregate along with metakaolin has not been initiated yet.”
Comment #2. The introduction requires more improvement by including more background about the partial substitution of various wastes supported by more recent literatures such as biological waste polymeric waste. These two types of wastes could be useful in current study; 10.1016/j.pnucene.2020.103285 and 10.1016/j.cscm.2021.e00664
Author’s response: The authors highly appreciate the reviwer’s comment to improve the manuscript. The authors have revised the introduction section and cited more recent literature to provide a comprehensive background of recent developments. The authors have also reviewed those two types of waste suggested by the reviewer. Please refer to lines 70-73, which are as follows:
“Biological waste (i.e., solidified plant) [30] and polymeric waste (i.e., polystyrene foam) [31] were also subjected to recent studies, resulting in positive outcomes and indicating that the search for novel materials can provide alternatives to conventional concrete materials.”
Comment #3. In current study, many parameters have been evaluated, what about the porosity of the produced composite?
Author’s response: The authors appreciate the reviewer's comment and agree that the outcome of porosity could better illustrate the mechanism of concrete's mechanical properties as the porosity increase with the decreased strength of the composites. The author regrets that the porosity of the composite was not studied and can be considered for the upcoming research work on the bond strength and durability properties after exposure to aggressive environments of concrete fabricated with glass aggregate. Here the authors emphasized the workability, destructive mechanical strengths, and the cost and carbon emission of the produced composite. Thanks
Comment #4. Spectroscopic investigations such as FTIR, XRD, SEM could add more important information about the proposed mortar.
Author’s response: The authors agreed with the reviewer’s comment that FTIR, XRD, and SEM could add more importance to the proposed mortar's microstructural and physicochemical characterization. The authors are undertaking another research project to investigate the durability, non-destructive strengths, microstructural and physicochemical characterization of glass aggregate composites. However, for the present study, these remain out of the scope.
Comment #5. The discussion needs more deep investigations.
Author’s response: The authors highly appreciate the reviewer’s comment to improve the manuscript. The authors have revised the manuscript as per the reviewer’s suggestions by identifying where deep investigations are needed and adding mechanisms supported by proper references to discuss the results. For example, please refer to lines 422-433, which are as follows:
“As mentioned in Section 3.2.1, the flat and flaky shaped GC can affect the matrix positively by interlocking and providing load transfer pathways. For this reason, incorporating GC at a lower percentage of 10% exhibited slightly higher flexural strength. However, when the GC content is further increased, this advantage can turn into a disadvantage because the overuse of GC can impart inhomogeneity in their orientation [70]. Also, the smooth surface of GC can create poor and porous ITZ between aggregates and cement paste [71]. The combined effect of these can significantly decrease the flexural strength of hardened concrete, when GC is employed.”
Also, the authors have added an ANOVA test for the concrete mixtures’ compressive strengths with respect to their glass cullet and metakaolin percentage. This analysis can be found in Section “3.2.3 Analysis of variance (ANOVA) for compressive strength test results” in the revised manuscript.
Comment #6. What's the research significance of this paper? What new insights do you provide? The authors are invited to provide in details.
Author’s response: Acknowledged. In recent years, it has been well established that glass particles or dust can be suitably employed to replace the fine aggregate in concrete partially. Similarly, metakaolin is also a known supplementary cementitious material. However, the authors have identified a gap in knowledge regarding glass aggregate as coarse aggregate and the combined effect of glass aggregate and metakaolin. This inspired the authors to conduct this study in isolated and combined ways. These statements are given in Section 1.1.
As reported in the revised manuscript, these outcomes indicate that only replacing coarse aggregate with glass cullet does not enhance the mechanical strength of concrete if used more than 10%. Contrastingly, if 10% metakaolin is employed along with glass cullet, concrete can experience enhanced mechanical strength up to 25% waste glass aggregate incorporation, which is the most significant insight of this paper. Also reported in the revised manuscript is that this combined approach can improve the cost and carbon emission efficiency of concrete mixtures along with improved fresh qualities, which can contribute to the ongoing search for sustainable concrete materials.
Comment #7. In conclusion, I believe that the theme of this manuscript can be consistent generally with the theme of Sustainability and specially with the SI: Recent Developments on the Use of Sustainable Retrofitting and Construction Materials.
Author’s response: The authors would like to thank the reviewer for this positive comment.
Comment #8. At the same time, the manuscript needs to be improved and the authors should edit the manuscript in accordance with the guidelines mentioned above.
Author’s response: The authors have carefully amended the manuscript considering all the comments by the reviewer and provided detailed explanations. The authors hope that the reviewer will be satisfied based on the amendments.

Round 2
Reviewer 2 Report
The manuscript can be accepted.
Reviewer 3 Report
The manuscript has been well revised, and it is recommended to be accepted for publication.
Reviewer 4 Report
Accepted in current from